# Distinct diet-microbiome associations in autism spectrum disorder

Yuqi Wu[1,2,9], Oscar Wong [3,4,9], Sizhe Chen [1,2], Yun Wang[1,2], Wenqi Lu[1,2], Chun Pan Cheung[1,2], Jessica Y. L. Ching [1,2], Pui Kuan Cheong[1,2], Sandra Chan[3,4], Patrick Leung [4,5], Francis K. L. Chan[1,4,6,7], Qi Su [1,2] ✉ & Siew C. Ng [1,2,7,8] ✉

Autism spectrum disorder (ASD) is linked to both altered gut microbiota and unhealthy diets; however, the mechanistic connections remain elusive. In this study, we conducted a systematic analysis of fecal microbiome metagenomic data, paired with granular dietary assessments and phenotypic profiles, across a cohort of 818 children (462 with ASD, 356 without ASD; mean age = 8.4 years; 27.3% female). By integrating dietary indices, nutrient intake, and food additive exposures, we uncovered ASD-specific linkages to the microbiome. Poor dietary quality correlated with aggregated core autistic symptoms, gastro-intestinal complications, and atypical eating behaviors. Notably, children with ASD exhibited a more pronounced diet-microbiome interaction network compared to neurotypical peers, suggesting heightened microbial sensitivity to nutritional inputs. Furthermore, synthetic emulsifiers—specifically polysorbate-80 and carrageenan—were associated with disrupted microbial connectivity in ASD, a phenomenon attenuated in neurotypical children. Our findings elucidate the mechanistic links between dietary factors—particularly synthetic food additives—and microbiome dysregulation in ASD, urging a re-evaluation of dietary guidelines for ASD populations and laying the ground-work for personalized nutritional strategies.

Autism spectrum disorder (ASD) is a neurodevelopmental condition marked by social communication challenges and restricted, repetitive behaviors[1]. Compared to neurotypical peers, children with ASD face a fivefold higher risk of feeding problems such as extreme food selectivity[2], as well as a much higher rate of gastrointestinal (GI) disorders[3–5]. These issues may interact bidirectionally with core ASD traits (e.g., sensory sensitivities) and gut microbiome imbalances, creating a cycle where diet, microbiota, and neurodevelopment influence one another[6,7]. Studies have identified distinct microbial patterns in ASD, including shifts in *Bacteroidetes* and *Clostridium* species, as well as functional changes in pathways related to inflammation and neurotransmitter balance[3,8]. While these findings have sparked interest in microbiome-targeted therapies for ASD[8,9], critical questions remain about how diet and gut microbes interact to shape neurodevelopmental outcomes.

Diet and medication are among the most powerful modulators of the gut microbiome[10,11]. In ASD, where diets are often limited (e.g., low in fiber, high in processed foods) and medications like antibiotics or

¹Microbiota I-Center (MagIC), Hong Kong SAR, China. ²Department of Medicine and Therapeutics, The Chinese University of Hong Kong, Hong Kong SAR, China. ³Department of Psychiatry, The Chinese University of Hong Kong, Hong Kong SAR, China. ⁴The D.H. Chen Foundation Hub of Advanced Technology for Child Health (HATCH), The Chinese University of Hong Kong, Hong Kong SAR, China. ⁵Department of Psychology, The Chinese University of Hong Kong, Hong Kong SAR, China. ⁶Centre for Gut Microbiota Research, The Chinese University of Hong Kong, Hong Kong SAR, China. ⁷Li Ka Shing Institute of Health Sciences, State Key Laboratory of Digestive Disease, Institute of Digestive Disease, The Chinese University of Hong Kong, Hong Kong SAR, China. ⁸New Cornerstone Science Laboratory, The Chinese University of Hong Kong, Hong Kong SAR, China. ⁹These authors contributed equally: Yuqi Wu, Oscar Wong. ✉e-mail: qisu@cuhk.edu.hk; siewchienng@cuhk.edu.hk

psychotropics are commonly used, these factors may uniquely disrupt microbial balance[12,13]. However, the role of specific dietary components—such as synthetic additives (e.g., emulsifiers) or nutrient deficiencies—in driving ASD-associated dysbiosis remains poorly understood[3,7]. This gap hinders the creation of evidence-based nutritional guidelines for ASD, a population with highly individualized food preferences and microbiome profiles. Personalized approaches that account for diet-microbiome interactions are critical to advancing tailored interventions.

Here, we examine how dietary habits and medications influence the gut microbiome in 818 children with and without ASD by incorporating fecal metagenomics, detailed dietary surveys, and clinical data. Our findings elucidate ASD-specific links between diet and microbial dysbiosis, paving the way for microbiome-aware dietary guidance to improve gut health and mitigate behavioral challenges in children with ASD.

## Results

### Diet-driven gut microbial signatures and phenotypic correlates in ASD

Given the relevance of the Chinese Children's Healthy Dietary Index (CCDI) for characterizing general dietary quality in our study population, we first conducted dietary profiling in a cohort of 818 children (mean age = 8.4 years; 27.3% female; children without ASD [non-ASD] = 356, ASD = 462, Fig. 1A), categorized by tertiles of CCDI. The analysis revealed that individuals with poorer dietary patterns exhibited more severe autistic symptoms, increased medication use, GI complications, and challenges across multiple eating behaviors, including desire to drink, food enjoyment, fussiness, and satiety responsiveness (all $p < 0.05$, Table 1). Further dietary comparisons identified nutritional insufficiency and low dietary quality across multiple metrics, including the Alternative Healthy Eating Index (AHEI), Dietary Inflammatory Index (DII), sulfur-diet score, and healthy food diversity (HFD) index, alongside elevated polysorbate-80 exposure, all of which correlated with diminished CCDI scores (Supplementary Table S1).

Consistent with prior studies, gut microbiome alpha diversity (Shannon index: coef = 0.07, standard error [SE] = 0.03, $p = 0.024$; observed richness: coef = 9.05, SE = 3.49, $p = 0.003$) and dysbiosis score (Dissimilarities-based evaluation as detailed in the "Methods", coef = −0.02, SE = 0.004, $p < 0.001$) differed significantly between children with ASD and those without (Fig. 1B, Supplementary Fig. 1A). A key finding was that the CCDI, among various continuous dietary indices, had a robust association with microbial dysbiosis in the ASD group ($\rho$ for Spearman correlation = −0.121, $p = 0.009$; t [degree of freedom = 461] = −0.258 for linear regression model adjusted by age, sex, GI conditions, and autistic symptoms, $\beta = −0.0004$, $p = 0.042$). In contrast, we did not observe this association in non-autistic peers (Fig. 1C, Supplementary Tables S2, 3 and Supplementary Fig. 1B, C). While the effect modification by ASD status did not attain statistical significance, the consistent pattern of association was verified in sensitivity analysis that excluded samples in accordance with the size and features of non-ASD (Supplementary Tables S2, 3).

To address the multidimensionality of eating behaviors, we derived principal components (PC1 and PC2) from the Children's Eating Behavior Questionnaire (CEBQ) subscales, which collectively explained 60.9% of variance (PC1: 33.1%, PC2: 27.8%; Supplementary Fig. 1D), and adjusted for these in subsequent models. To quantify the influence of host factors on gut microbiota composition, we performed a multivariate PERMANOVA analysis through Bray-Curtis distances. Among phenotypic factors, dietary quality emerged as a key contributor to microbial taxonomic variation, explaining 1.57% of variance ($p < 0.01$)—second only to stool consistency (Bristol Stool Chart) and ahead of age, emulsifier exposure, and BMI (Fig. 1D). Medication use, in contrast, minimally influenced microbial

composition, aligning with its negligible impact on functional pathway variance (Supplementary Fig. 1E). While effect sizes for these factors were comparable in children without ASD, most associations lost statistical significance, underscoring the ASD-specific interplay between diet, phenotype, and gut ecology (Fig. 1D).

### ASD-specific diet-microbiome interactions

Building on the observed connections between diet, food additives, and microbial ecology, we systematically evaluated associations between dietary factors (indices, components, and food groups) and gut microbiome profiles within the ASD population. These associations were analyzed using MaAsLin2, adjusted for the age, sex, and GI conditions that were identified above. Significant microbial features were selected by ranking them based on the number of significant associations across the metadata panels, with the top 50 unique features retained for each panel. The coefficients, adjusted q-values, and prevalence for each microbial species are detailed in Supplementary Data 1, with statistical descriptions provided in the Description of Additional Supplementary Files. These microbial species (Fig. 2A), along with MetaCyc-based functional pathways and KEGG orthologs (KO) (Supplementary Figs. 2, 3), revealed distinct diet-microbe associations comping the non-ASD peers when conducting the same analysis procedure. Specifically, *Firmicutes SGB4348*, *Megamonas funiformis*, and *Lacrimispora amygdalina* emerged as indicators of healthier dietary patterns in ASD, characterized by higher CCDI scores (*F.SGB4348*: coef = 0.64, SE = 0.19, qval = 0.01 in ASD vs. coef = 0.18, SE = 0.28, qval = 0.79 in non-ASD; *M.funiformis:* coef = 0.56, SE = 0.17, qval = 0.01 in ASD vs. coef = 0.24, SE = 0.31, qval = 0.91 in non-ASD; *L. amygdalina:* coef = 0.72, SE = 0.19, qval = 0.01 in ASD vs. coef = 0.01, SE = 0.28, qval = 0.99 in non-ASD) and lower DII scores (*F.SGB4348*: coef = −0.49, SE = 0.19, qval = 0.09 in ASD vs. coef = −0.32, SE = 0.29, qval = 0.84 in non-ASD; *M.funiformis:* coef = −0.41, SE = 0.17, qval = 0.10 in ASD vs. coef = −0.38, SE = 0.25, qval = 0.51 in non-ASD; *L. amygdalina:* coef = −0.65, SE = 0.18, qval = 0.01 in ASD vs. coef = 0.24, SE = 0.23, qval = 0.69 in non-ASD). These associations were replicated in the ASD group using a sample-size matched sensitivity analysis (Supplementary Data 2). Consistently, the population-wide regression model with an interaction term (diet × ASD) indicated the effect modification of ASD status on dietary associations with the abundance of *L. amygdalina* and *Anaerostipes hadrus* (qval < 0.2, Supplementary Fig. 4, Supplementary Data 3). Notably, decreased abundance of *L. amygdalina* was specifically correlated with insufficient vitamin and mineral intake in the ASD group (|coef| > 0.5 and qval < 0.1), while *Massilioclostridium coli* and *Escherichia coli* exhibited inverse correlations with these dietary metrics (Supplementary Data 3). While these associations were modest overall, they were largely absent in non-ASD peers, and Venn analyses confirmed no overlap in diet-microbiome interaction patterns between groups, underscoring the ASD-specific nature of these relationships (Supplementary Fig. 5A, Supplementary Data 4).

The breadth of diet-microbe associations varied by dietary variable, showing stronger connections in ASD compared to their non-ASD peers, who exhibited negligible linkages. The dietary factors exhibiting the greatest magnitude of associations with microbial species—across nutritional components, dietary indices, and food groups—were protein intake (28 significant associations in total), CCDI scores (27 in total), and bean consumption (18 in total) (Fig. 2B, Supplementary Fig. 5B). By looking into specific nutrients in relation to microbial shifts, starch and magnesium emerged as the nutrients with the second-highest number of significant associations after protein intake, exhibiting 27 and 20 significant associations, respectively (Fig. 2B). Consistent results were observed at the compositional level in diet-microbiome associations measured using two complementary machine-learning approaches, each trained on relative microbial abundances to predict each dietary variable quantified by correlation

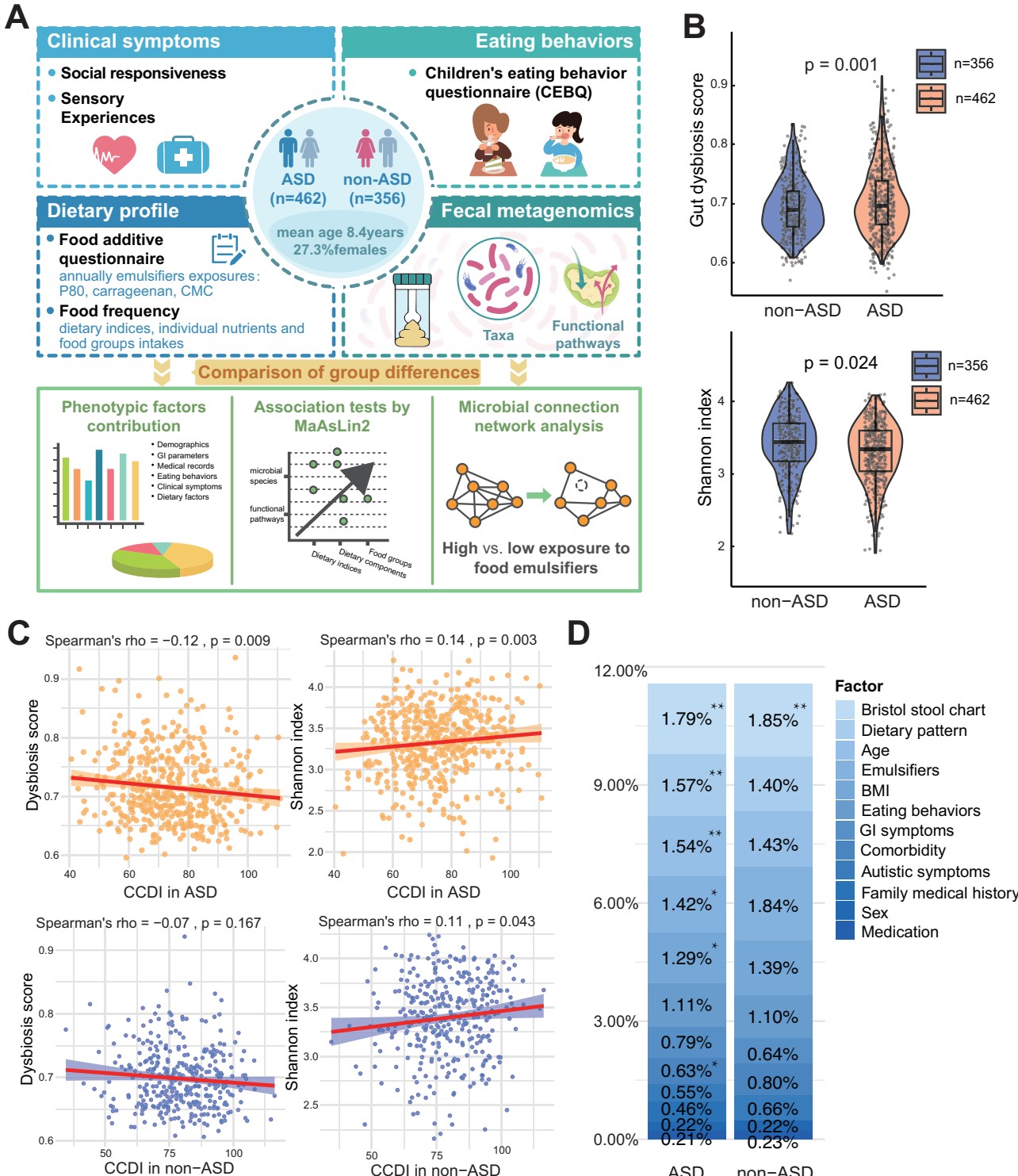

**Fig. 1 | Diet-driven gut microbial signatures and phenotypic correlates in ASD.**
**A** Study design overview. Schematic representation of the analytical framework integrating dietary profiling, microbial compositions, and phenotypic characterization. The images were created by Y.Wu without third-party resources involved. **B** Microbial community alterations in gut microbial dysbiosis and microbial diversity score between children with and without ASD. Statistical significance was assessed using a two-tailed Mann-Whitney *U*-test comparison. **C** The effect size of diet and other confounders on the variability of the gut microbiome community is examined. The proportion of gut microbial variation explained by various phenotype factors was assessed using PERMANOVA. Bray-Curtis distance matrices, based on the relative abundances of microbial species, were analyzed using the adonis function in the R package vegan with 9999 permutations. **D** The associations between a healthy dietary pattern (CCDI) and gut microbial features vary between children with and without ASD. 95% confidence interval shown as a shaded in the scatter plot.

**Table 1 | Demographic summary of participants in the study**

|  | CCDI (Tertile 1) N = 273 | CCDI (Tertile 2) N = 273 | CCDI (Tertile 3) N = 272 | p-value |
|---|---|---|---|---|
| Age | 8.00 (7.00, 10.00) | 8.00 (7.00, 10.00) | 9.00 (7.00, 10.00) | 0.130 |
| Male | 213 (78%) | 191 (70%) | 191 (70%) | 0.056 |
| ASD diagnosis | 183 (67%) | 152 (56%) | 127 (47%) | <0.001 |
| BMI | 16.1 (14.7, 18.6) | 16.0 (14.5, 18.6) | 16.2 (15.1, 18.4) | 0.700 |
| Family medical history | 38 (13.9%) | 39 (14.3%) | 35 (12.8%) | 0.475 |
| Comorbidity | 88 (32%) | 95 (35%) | 76 (28%) | 0.200 |
| Medication use | 51 (19%) | 54 (20%) | 45 (16%) | <0.001 |
| GI symptoms | 56 (21%) | 29 (11%) | 26 (9.5%) | <0.001 |
| Bristol stool chart |  |  |  | <0.001 |
| Type 1, 2 | 65 (24.0%) | 44 (15.8%) | 39 (14.3%) |  |
| Type 3, 4 | 147 (54%) | 166 (61%) | 177 (65%) |  |
| Type 5, 6, 7 | 26 (9.6%) | 22 (8.1%) | 23 (8.5%) |  |
| Autistic symptoms |  |  |  |  |
| SEQ-hyper | 2.00 (1.64, 2.43) | 1.86 (1.50, 2.29) | 1.71 (1.50, 2.07) | <0.001 |
| SEQ-hypo | 1.50 (1.17, 2.00) | 1.50 (1.17, 2.00) | 1.33 (1.17, 1.83) | 0.048 |
| SRS-RRB | 56 (48, 66) | 54 (46, 66) | 50 (45, 62) | 0.002 |
| SRS-SCI | 63 (55, 73) | 61 (54, 71) | 59 (52, 66) | <0.001 |
| Eating behaviors (CEBQ) |  |  |  |  |
| DD | 2.67 (2.00, 3.33) | 2.67 (2.00, 3.33) | 2.33 (2.00, 3.00) | 0.025 |
| EF | 3.00 (2.50, 3.75) | 3.25 (2.75, 3.75) | 3.50 (2.75, 4.00) | <0.001 |
| EOE | 1.75 (1.25, 2.25) | 1.75 (1.25, 2.25) | 1.75 (1.25, 2.00) | 0.600 |
| EUE | 2.75 (2.25, 3.25) | 2.75 (2.25, 3.25) | 2.50 (2.00, 3.25) | 0.600 |
| FF | 1.50 (1.17, 1.83) | 1.33 (1.00, 1.67) | 1.17 (1.00, 1.50) | <0.001 |
| FR | 2.40 (2.00, 2.80) | 2.40 (2.00, 2.80) | 2.20 (2.00, 2.80) | 0.200 |
| SE | 2.25 (1.50, 2.75) | 2.00 (1.63, 2.75) | 2.00 (1.50, 2.75) | 0.990 |
| SR | 2.00 (1.80, 2.60) | 2.00 (1.60, 2.40) | 2.00 (1.60, 2.40) | 0.044 |

The study populations were categorized into tertiles of the Chinese Children Healthy Dietary Index (CCDI), indicating a less healthy dietary pattern with an average score of 63 (interquartile range [IQR]: 57, 67), a moderate quality with an average score of 76 (IQR: 73, 80), and a healthier pattern with an average score of 91 (IQR: 86, 96). Group comparisons were performed using Kruskal-Wallis rank sum tests for continuous variables, and Pearson's Chi-squared tests were used for categorical variables. All reported exact *p*-values are two-tailed testing. *BMI* body mass index, *GI* gastrointestinal, *CEBQ* the Children's Eating Behavior Questionnaire, *DD* desire for drinks, *EF* enjoyment of food, *EOE* emotional overeating, *EUE* emotional undereating, *FF* food fussiness, *FR* food responsiveness, *SE* slowness in eating, *SR* satiety responsiveness.

coefficients (ρ) for regression and area-under-the-curves (AUCs) for classification ("Methods"). The random forest regression models demonstrated diet-associated species outperformed functional pathway-based predictions across 100 training/testing folds, where strong concordance was observed between predicted and actual values for protein intake (ρ > 0.35) and CCDI scores (Fig. 2B, Supplementary Fig. 5C). The random forest-based binary classification further achieved higher AUC performance in children with ASD compared to non-ASD (Fig. 2B). Notably, emulsifiers—particularly polysorbate-80 and carrageenan—demonstrated stronger associations with microbial features in the ASD group (Fig. 2B, Supplementary Fig. 6). Overall, we demonstrated that children with ASD exhibit distinct and stronger diet-gut microbiome associations compared to neurotypical peers.

**Magnesium/protein enrichment and starch depletion in ASD**

Given the prominence of microbial associations with CCDI scores and key nutrients (protein, starch, magnesium), we investigated gut microbial shifts in ASD children stratified by high intake of these dietary components. Notably, *M. funiformis* was enriched in ASD with elevated CCDI scores, high magnesium/protein intake, and reduced starch consumption, while four species—*Cibionibacter quicibialis*, *Collinsella aerofaciens*, *Bacteroides nordii*, and *Vescimonas coprocola*—were depleted in those with high starch but low protein intake (Fig. 3A). These trends attenuated in neurotypical peers

(Supplementary Fig. 7A–D, Supplementary Data 5), with minimal overlap in microbial responses—a contrast that persisted against a sample-size-matched ASD cohort (Supplementary Data 6). The ASD-specific increase of *M. funiformis* in response to healthy dietary patterns collectively demonstrated the diet-by-ASD interaction (p for interaction < 0.05, Supplementary Data 6). Functional pathway analyses revealed reduced activity in Dihydroxy-6-naphthoate biosynthesis linked to high starch intake and suppressed fatty acid biosynthesis initiation (*E. coli*) associated with lower CCDI scores and magnesium intake (Fig. 3B, Supplementary Data 7). Concurrently, gene-set enrichment analysis based on KEGG orthology (KO) indicated enhanced microbial processing functions related to amino acid and cofactor biosynthesis (Supplementary Fig. 7E), suggesting a state of metabolic dysregulation in children with ASD linked to dietary imbalances. Collectively, these diet-microbe-functional perturbations highlight mechanistic pathways that may exacerbate challenges in ASD populations with suboptimal nutritional patterns.

To elucidate the role of microbiome in mediating these diet metrics associations in ASD, we performed mediation analyses. Among the specific nutritional components examined, *Faecalibacterium prausnitzii* exerted a beneficial mediating influence on the associations between protein/zinc intake and social communication (average causal mediation effect [ACME, 95% CI] = −0.125 [−0.245, −0.027], p for mediation = 0.006, proportion = 24.6%) and hyposensitivity (ACME =

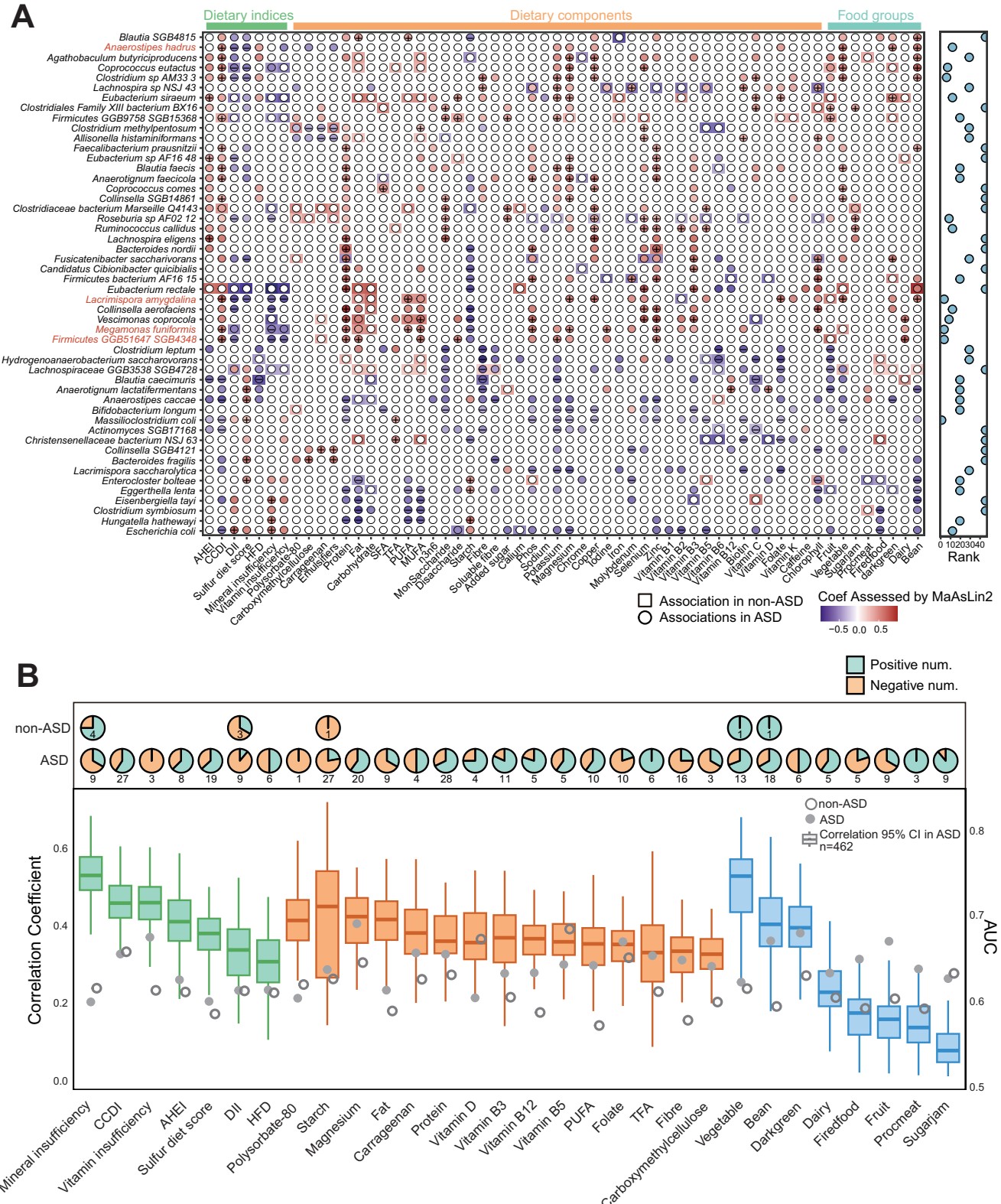

**A**

**B**

−0.004 [−0.009, −0.0005], $p = 0.02$, 15.3%) (Supplementary Fig. 8, Supplementary Data 8). Notably, *Coprococcus eutactus* emerged as the most prominent mediator, exhibiting the highest proportion of effect for several diet-restricted and repetitive behaviors (RRB) associations, including CCDI (ACME = 0.013 [0.001, 0.029], $p = 0.02$, 15.8%), DII (ACME = −0.098 [0.001, 0.019], $p = 0.02$, 19.9%), and bean consumption (ACME = 0.009 [−0.210, −0.011], $p = 0.02$, 35.4%).

## Emulsifier-driven disruption of microbial ecological networks in ASD

Of particular interest were dietary emulsifiers, given their established potential to perturb gut microbiota and their notable contribution to the variance explained in microbial composition (Figs. 1D, 2B). *Ruminococcus sp AF13 28* emerged as a hallmark taxon inversely linked to polysorbate-80 and carrageenan exposure in ASD (Fig. 4A).

**Fig. 2 | ASD-specific microbial associations with dietary profiling. A** Heatmap displays the top 50 diet-associated microbial species in children with ASD (in circle shape) that lost significant associations in non-ASD peers (in rectangular shape). Significant associations (MaAsLin2, qval < 0.1, adjusted for age, sex, and gastrointestinal conditions) are color-coded by dietary category: green (dietary indices), orange (nutritional components), and blue (food groups). Coefficients details and prevalence for each species were provided in Supplementary Data 1, detailed in the Description of Additional Supplementary Files. The significant associations (qval < 0.05) are annotated with + or - according to the directions of associations. **B** Microbial predictors of dietary profiling. Sector diagrams quantify the prominence of significant diet-microbe associations (top: ASD; bottom: non-ASD).

Machine learning models demonstrate the predictive power of microbiome features by generating correlation results (left axis) and AUC values (right axis), respectively. Box plots display the agreement between actual dietary values and microbiome-based predictions from 10-fold cross-validated regression models in ASD, with error bars representing the 95% confidence interval (CI). Predictive performance is further quantified by median AUC values from binary classification (solid circles: ASD; hollow circles: non-ASD). CCDI, the Chinese Children's Healthy Dietary Index; AHEI, Alternative Healthy Eating Index; DII, Dietary Inflammatory Index; HFD, healthy food diversity index; SFA, saturated fatty acids; MUFA, monounsaturated fatty acids; PUFA, polyunsaturated fatty acids; TFA, trans-fatty acids.

Conversely, *Collinsella SGB4121* and *Bacteroides fragilis*—species scarcely represented in neurotypical peers—were enriched with high carrageenan intake. While emulsifier exposures showed limited broad associations with microbial composition. Characterization of population-wide emulsifier exposure tertiles of the cohort revealed demonstrated ASD-specific microbial network instability, evident from significant topological alterations (Table 2) and a positive association between emulsifier levels and dysbiosis score (Supplementary Fig. 1C). To ensure statistical robustness, we performed bootstrap analyses (n = 1000 iterations) for each network metric. In the highest exposure tertile, we observed fragmented microbial connectivity, marked by significant reductions in network degree, number of edges, and average degree (bias-corrected p < 0.01) (Fig. 4B, C, Table 2 and Supplementary Fig. 9A, B). The network connectivity, also as reflected by the clustering coefficient, was specifically decreased in the ASD group with higher polysorbate-80 intake (Table 2). Simulation attacks confirmed this fragility, as both random and targeted node removals disproportionately disrupted network integrity in ASD (Fig. 4D, E). We identified several taxa (e.g., *Enterocloster bolteae*, *Hungatella hathewayi*, and *Clostridium symbiosum*) as potential keystone species due to their high connecting degrees in the interaction network, displayed diminished interconnectivity in high-exposure ASD networks (Fig. 4B, C), while functional pathways critical for nutrient synthesis (e.g., erythronate and biotin biosynthesis) were suppressed (Supplementary Fig. 9C, D). KEGG enrichment analysis further revealed that polysorbate-80/carrageenan exposure concurrently downregulated the 2-oxocarboxylic acid metabolism and citrate cycle pathways (Supplementary Fig. 9E). Non-ASD children, in contrast, maintained stable network architecture and functional resilience across emulsifier exposure levels (Fig. 4F, G). These findings position synthetic emulsifiers as potent disruptors of microbial ecology in ASD, exacerbating network vulnerability and metabolic deficits already present in this population.

**Diet- and medication-independent microbial biomarkers in ASD**
These findings underscore the unique interplay between diet and gut ecology in ASD, prompting us to investigate whether dietary variables could influence the stability of previously identified microbial biomarkers for ASD diagnosis[8]. Reassuringly, ASD-linked microbial signatures—spanning taxonomic groups, functional pathways, and KO gene families—remained independent of dietary influences (Supplementary Fig. 10). Equally critical, systematic evaluation of medications (ADHD drugs, antipsychotics, melatonin, cyproheptadine) and nutritional supplements showed no meaningful interaction with these biomarkers in ASD models. This dual resilience to dietary and pharmacological confounders solidifies the clinical utility of gut microbial markers in ASD, offering diagnostically robust insights unmarred by lifestyle or treatment variables.

## Discussion
This study reveals critical insights into the interplay between dietary patterns, gut microbiome dynamics, and clinical features in children

with ASD. Our analyses demonstrate that children with ASD exhibit distinct diet-microbiome associations compared to neurotypical peers, characterized by (i) nutrient-specific microbial shifts, (ii) heightened sensitivity of gut microbial networks to synthetic emulsifiers, and (iii) disrupted functional pathways linked to neuroprotective metabolite synthesis. Our study shows how certain food factors may disrupt gut microbiota in ASD, calling for an urgent review of current dietary advice and paving the way for tailored nutrition plans.

Children with ASD frequently exhibit atypical eating behaviors—food selectivity, fussiness, and altered satiety responsiveness—that contribute to nutritionally imbalanced diets low in fiber and micronutrients but high in processed foods[14,15]. These behaviors, compounded by GI complications (e.g., constipation, diarrhea), create a cyclical relationship where poor dietary quality exacerbates GI distress, further altering gut microbiota composition[16]. Our data align with this paradigm: the Bristol Stool Scale, a proxy for GI health, explained the largest proportion of microbial variance, followed by dietary indices. This suggests that dietary interventions targeting both nutrient adequacy and GI symptoms management could break this cycle[10,17], though personalized approaches are essential given the heterogeneity of ASD phenotypes[18]. This critical question—regarding how heterogeneity in GI conditions modulates diet-microbiome interactions—prompted a focused sub-analysis within the ASD population, stratified by the presence of GI symptoms and irregular Bristol Stool types. It revealed that dietary indices exerted a more pronounced association with microbial features in ASD children without GI complications (Supplementary Fig. 11), suggesting that GI health appears to be a prerequisite for robust diet-microbiome interactions[19].

The gut microbiota of children with ASD displayed heightened responsiveness to dietary variables, with protein intake, CCDI scores, and bean consumption exerting the strongest effects on microbial composition. Notably, *Firmicutes* species associated with healthy dietary patterns (e.g., high CCDI scores) may mediate beneficial effects through fiber fermentation and short-chain fatty acid production, which modulate gut-brain signaling[20,21]. Complementary KEGG enrichment analysis linked CCDI-associated orthologs to cofactor biosynthesis (e.g., thiamine metabolism), suggesting its role in the improved ASD-related outcomes[22]. Conversely, high starch intake—common in ASD—correlated with reduced dihydroxy-6-naphthoate biosynthesis, a pathway critical for vitamin K2 synthesis[23,24]. Vitamin K2 deficiency has been implicated in neuroinflammation and mitochondrial dysfunction, both hallmarks of ASD, suggesting a mechanistic link between diet, microbial metabolism, and neurological outcomes[23,24]. Mediation analysis revealed that *Faecalibacterium prausnitzii* mediated the beneficial effects of protein and zinc intake on ASD symptoms, likely through its anti-inflammatory and butyrate-producing properties[25]. *Coprococcus eutactus* exhibited the strongest mediation effect, suggesting a key role in gut-brain communication and potential as a probiotic target for dietary interventions[26,27]. These findings highlight the possibility of microbiota-directed therapies for ASD.

Children with ASD exhibited pronounced ecological fragility in response to emulsifier exposure, marked by fragmented microbial

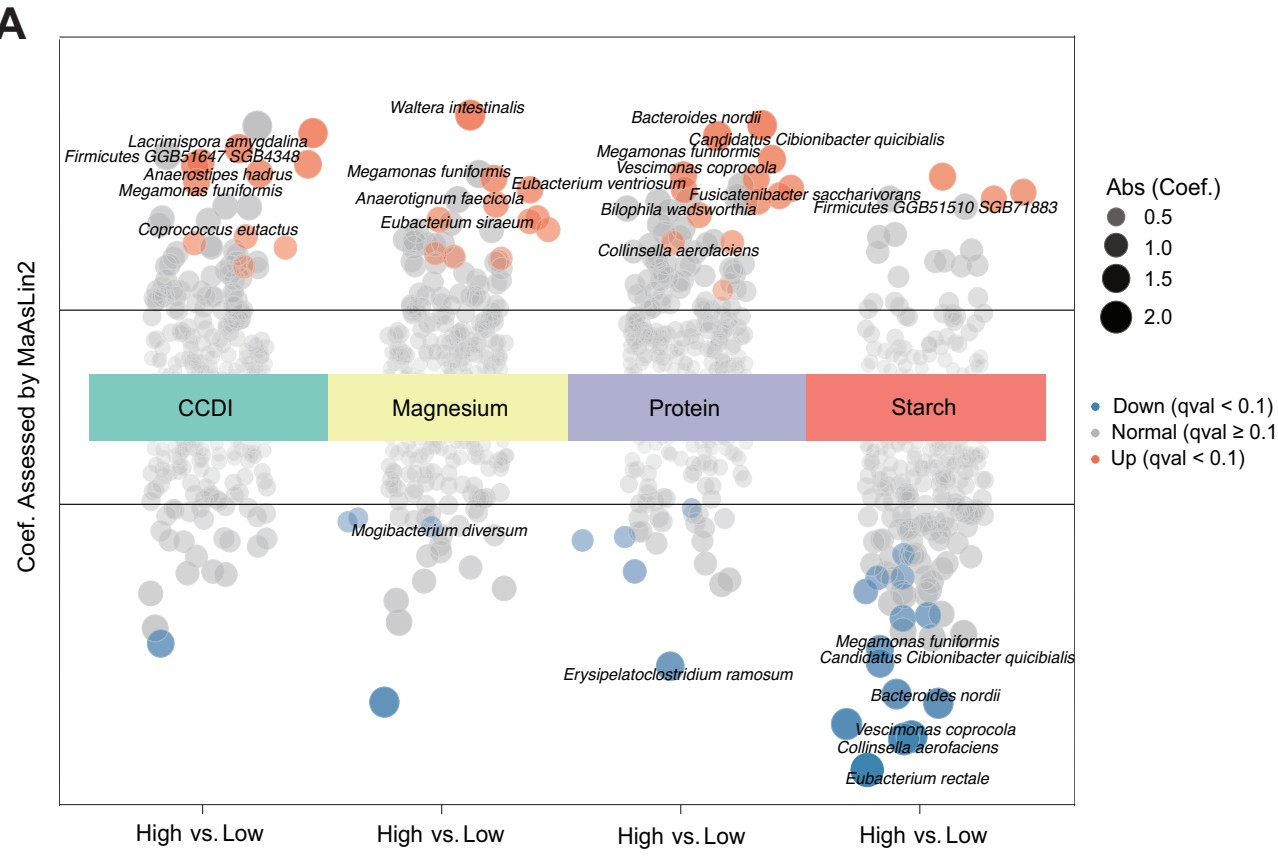

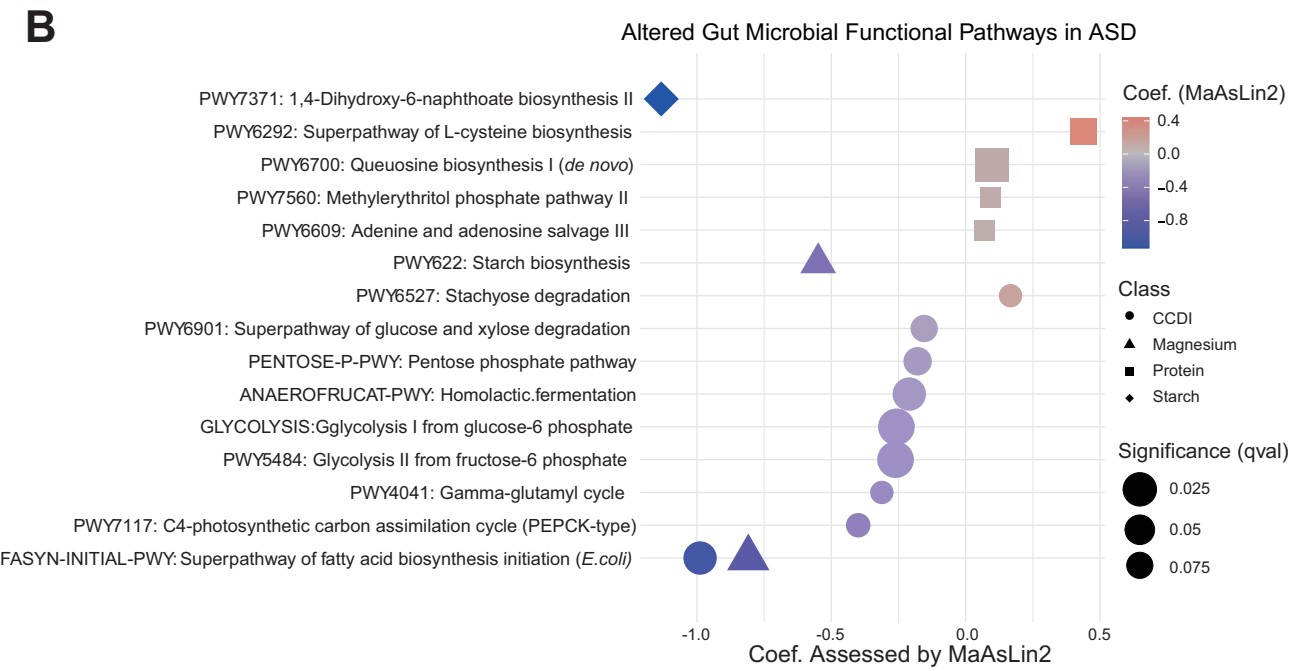

**Fig. 3 | Microbial and functional adaptations to specific dietary components in ASD. A** Species-level response to dietary factors. Differential abundance of gut microbiota associated with four key dietary factors in ASD, including the Chinese Children's Healthy Dietary Index (CCDI) scores, magnesium, protein, and starch intakes. Results show MaAsLin2 coefficients after adjustment for age, sex, and gastrointestinal conditions, with high vs low intake groups (reference) compared.

Significant associations (qval < 0.1) are color-coded (orange: enrichment in high intake; blue: depletion). Top hits (qval < 0.05) are annotated with species names. **B** Functional pathway modulation. Corresponding changes in microbial metabolic pathways (MetaCyc) for the same dietary factors. Analyzed pathways had >10% sample prevalence. Significance thresholds and visualization scheme match panel (**A**).

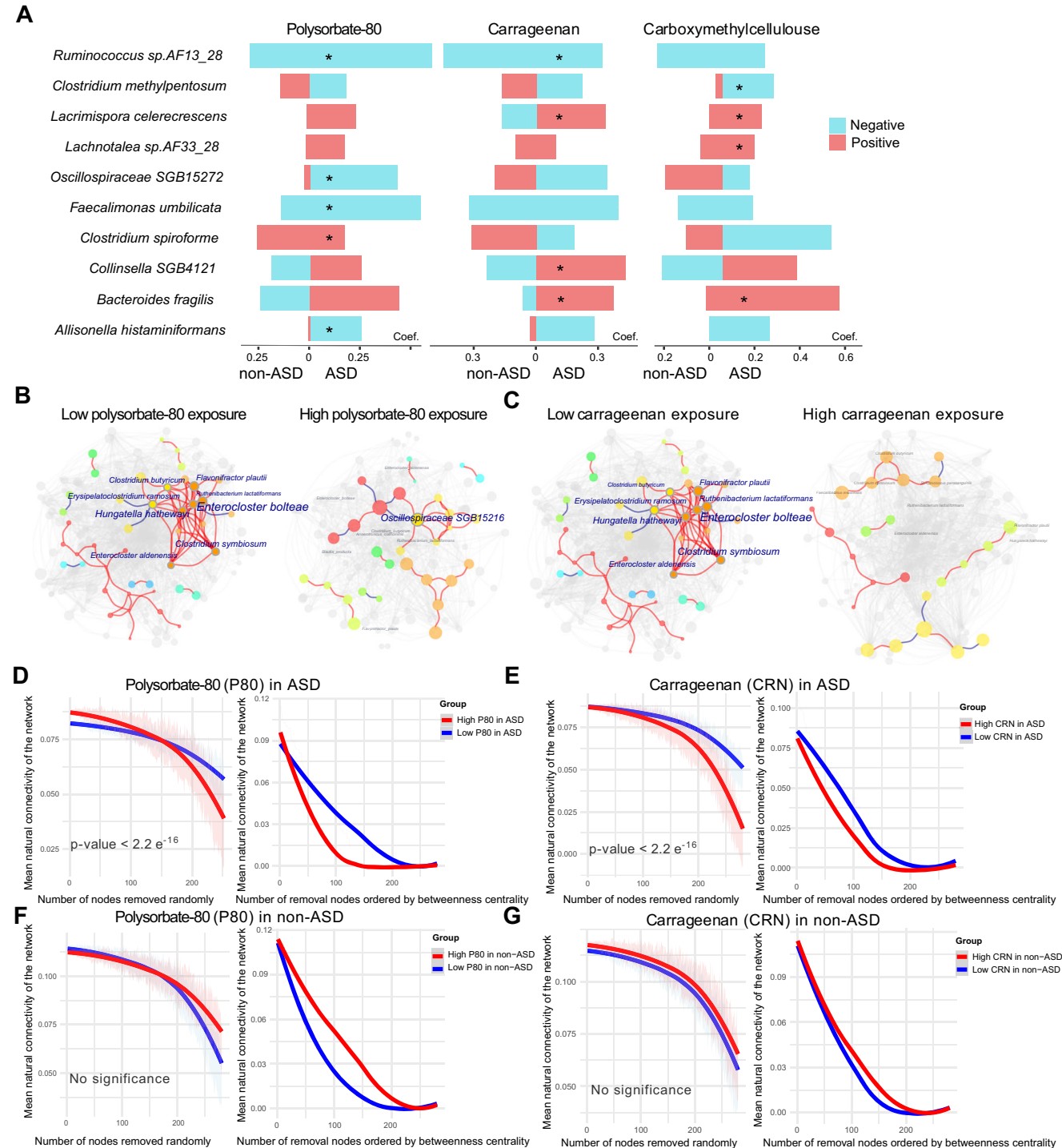

**Fig. 4 | Emulsifiers-driven disruption of microbial ecological networks in ASD.** **A** Differential abundance analysis (MaAsLin2, adjusted for age, sex, and gastro-intestinal conditions identified) reveals distinct microbial responses to dietary emulsifiers (polysorbate 80; carrageenan; carboxymethylcellulose) in ASD (right) versus non-ASD (left) peers. **B**, **C** Disrupted microbial connection network in ASD (right) rather than non-ASD (left) with high exposures to polysorbate-80 exposure (**B**) and carrageenan (**C**). **D**–**G** Stability of the remaining network was assessed in ASD under high and low polysorbate-80 exposure (**D**) and carrageenan exposure (**E**) through random attacks and node removal based on ranked degrees. We

simulated the influence of microbes' loss on the network by referring to the natural connectivity index. In addition to removing modes randomly (left panel), the order for selecting taxa in the co-occurrence networks was also performed based on the ranked degrees of nodes (right panel). The 95% confidence interval from the random node removal simulation, performed via the Hamiltonian Monte Carlo method, is represented as shaded around the curve plot. Group differences were assessed using the unpaired Wilcoxon rank-sum test (detailed in "Methods"). The same simulation attack assessments were conducted in the non-ASD group (**F**, **G**).

**Table 2 | The topological features of the microbial connection network according to the exposure levels of food emulsifiers**

| Polysorbate 80 | non-ASD | | | ASD | | |
|---|---|---|---|---|---|---|
| Parameters | Low (n = 143) | Moderate (n = 115) | High (n = 98) | Low (n = 130) | Moderate (n = 157) | High (n = 175) |
| Number of edges | 85[bc] | 111[bc] | 119[c] | 76[b] | 57[b] | 45[a] |
| Number of positive edges | 76[bc] | 105[bc] | 113[c] | 68[b] | 49[b] | 38[a] |
| Number of negative edges | 9[c] | 6[a] | 6[a] | 8[c] | 8[c] | 7[b] |
| Number of vertices | 59[b] | 70[c] | 75[c] | 47[a] | 53[a] | 39[a] |
| Average degree | 2.881[ab] | 3.171[ab] | 3.173[b] | 3.234[b] | 2.151[ab] | 2.307[a] |
| Average path length | 3.658[a] | 3.679[a] | 5.794[a] | 4.303[a] | 2.425[a] | 2.507[a] |
| Clustering coefficient | 0.426[ab] | 0.458[ab] | 0.428[ab] | 0.558[b] | 0.333[a] | 0.361[a] |
| Centralization degree | 7.241[a] | 7.942[a] | 8.946[a] | 7.934[a] | 8[a] | 7.894[a] |
| Modularity (> 0.4) | 0.626[a] | 0.614[a] | 0.584[a] | 0.45[a] | 0.736[a] | 0.626[a] |
| Number of modules | 11[ab] | 13[ab] | 14[ab] | 10[ab] | 14[b] | 9[a] |
| Carrageenan | non-ASD | | | ASD | | |
| Parameters | Low (n = 142) | Moderate (n = 132) | High (n = 82) | Low (n = 131) | Moderate (n = 140) | High (n = 191) |
| Number of edges | 67[bc] | 119[bc] | 79[c] | 69[b] | 70[bc] | 32[a] |
| Number of positive edges | 64[bc] | 107[bc] | 74[c] | 64[b] | 64[bc] | 24[a] |
| Number of negative edges | 3[a] | 12[c] | 5[b] | 5[b] | 6[b] | 8[b] |
| Number of vertices | 51[b] | 73[bd] | 57[c] | 43[b] | 61[b] | 31[a] |
| Average degree | 2.627[bc] | 3.26[c] | 2.772[c] | 3.209[b] | 2.295[b] | 2.064[a] |
| Average path length | 4.965[a] | 5.212[a] | 2.193[a] | 2.173[a] | 2.761[a] | 2.799[a] |
| Clustering coefficient | 0.574[a] | 0.47[a] | 0.672[a] | 0.594[a] | 0.275[a] | 0.328[a] |
| Centralization degree | 6.441[a] | 9.875[a] | 7.357[a] | 9[a] | 5.8[a] | 5.1[a] |
| Modularity (> 0.4) | 0.684[a] | 0.637[a] | 0.672[b] | 0.506[a] | 0.738[ab] | 0.658[a] |
| Number of modules | 10[ab] | 13[ab] | 15[c] | 8[b] | 12[a] | 7[a] |

In the absence of standard limits for annual intake of food emulsifiers, we classified the general population into tertiles, categorized as low, moderate, and high exposure groups. The topological features of the microbial connection network were analyzed separately in children with and without ASD using the igraphR package. To estimate the variability of network metrics, we employed a nonparametric bootstrap procedure with 1000 iterations via repeated resampling with replacement. Two-tailed, bias-corrected p-values were computed for the statistical differences of each topological feature. The compact letter (a, b, c) display denoted significant differences between groups, with those not sharing any common letter determined as significantly different after multiple comparisons ($p < 0.05$).

networks and loss of keystone taxa (*Ruminococcus sp AF13 28, Enterocloster bolteae*)[28]. The significant loss of alpha diversity and the exacerbated dysbiosis associated with increased exposure to carrageenan/polysorbate-80 in autistic children (Supplementary Fig. 1C) offered a plausible mechanism for the pronounced network instability observed specifically in ASD following the dietary disturbance. These disruptions mirror findings in inflammatory bowel disease, where emulsifiers compromise mucus barrier integrity and promote dysbiosis[29,30]. In ASD, such perturbations may amplify existing GI and neuroimmune dysfunction, creating a feedforward loop of microbial instability[3,31]. The outsized effect of carrageenan/polysorbate-80—achieving prediction metrics ($\rho = 0.4$, AUC = 0.65) rivaling clinical biomarkers—calls for urgent scrutiny of their safety in ASD populations.

This study's strengths include granular profiling of diet (indices, additives, nutrients) and multi-omics microbiome analysis, along with a comprehensive consideration of multiple variables, such as GI complications and behavioral challenges[3,11]. However, limitations warrant caution: (i) The cross-sectional design precludes causal inference, and the discrepant sample sizes between ASD and non-ASD groups may influence statistics. Nevertheless, sensitivity analyses using matched subgroup sizes consistently confirmed the distinct diet-microbiome associations observed in ASD and, meanwhile, cemented an interaction model to support an ASD-specific microbial response. The absence of neurodevelopmental and psychiatric comorbidities in the non-ASD group minimized the heterogeneity of the population and solidified our non-significant findings (Supplementary Table S4). (ii) Dietary data, though granular, were self-reported. Despite the data variability of diet being sufficient for the primary analyses, the limited range in certain micronutrients (e.g., B vitamins) may have reduced sensitivity for detecting subtler associations (Supplementary Table S4). As dietary habits are culturally embedded, our findings require validation in more diverse populations to ensure generalizability. Meanwhile, our findings positioned GI health as a potential effect modifier of diet-microbiome interactions, necessitating careful consideration in subgroups to avoid biased interpretations. (iii) Emulsifier effects may be compounded by unmeasured additives, and dietary heavy metals, as a risk factor, may also interact with the gut microbiome. Future studies specifically designed to incorporate biomarker-based assessments or utilize detailed dietary databases with comprehensive additive composition are urgently needed to elucidate the role of these substances. (iv) Finally, microbial network analyses warrant mechanistic explorations into microbiome-directed signals and gene-enrichment functions, ultimately requiring validation in gnotobiotic models.

Overall, our findings position the ASD gut microbiome as a factor both engaged in an interplay with diet and associated with dietary responses, with implications for precision nutrition. The stability of ASD-specific microbial biomarkers supports their diagnostic utility[32], while the additive-driven ecological fragility highlights actionable targets for dietary guidelines. Whether those ASD-specific biomarkers influence the overall microbiome community niche, which may in turn affect microbiome-diet interactions, opens up exciting avenues for exploration. Future work should prioritize longitudinal studies to unravel causality and clinical trials testing emulsifier-restricted, microbiota-targeted diets. By integrating dietary management with microbial modulation, we may mitigate GI comorbidities and improve quality of life in ASD.

## Methods

### Study cohort and descriptions

We initiated a prospective case-control study in Hong Kong, aiming at investigating the correlation between gut microbiota composition and autism symptoms and severity over time, as well as the validation of previously identified microbial markers of ASD, considering genetics and other confounding factors, including ASD-related symptoms and dietary information. Children with ASD were diagnosed by psychiatrists according to the fifth edition of the Diagnostic and Statistical Manual of Mental Disorders (DSM-5) and were recruited from the Child and Adolescent Psychiatric Clinic of the New Territory East Cluster (NTEC) of the Hospital Authority in Hong Kong from December 2021 to December 2023. Non-autistic children, without first-degree relatives diagnosed with autism, by screening negative on the Autism-Spectrum Quotient-10 (AQ-10) and absence of psychiatric disorder according to the DISC-5, were recruited from the community during the same period. Both groups included children under the age of 12, alongside exclusion criteria that comprised a known history of intellectual disability, psychosis, depression, and neurological disorders. No statistical method was used to predetermine sample size. Demographic details and several questionnaires used for the assessments of the severity of autistic symptoms, GI conditions, medication history, and dietary intake, were obtained from parents' report[8]. Biological measurements were also conducted at the baseline. A total of 818 children (ASD = 462 and non-ASD = 356, 27.3% females, age range: 3–12 years) of Chinese ethnicity, with complete dietary and gut metagenomics data, were included in the current study. The non-ASD group exhibited no abnormal behaviors following ASD screening and did not have any comorbidities. The study protocol adhered to the principles outlined in the Declaration of Helsinki and received ethics approval from the Joint Committee on Clinical Research Ethics, CUHK-New Territories East Hospital Cluster (CUHK-NTEC CRE, Ref.: 2021.550). Written informed consent was obtained from the parents or caregivers of the participants.

### Dietary questionnaires and assessments

To comprehensively evaluate dietary associations with the gut microbiome, we treated multiple dietary metrics as continuous measures, including indices (CCDI, AHEI, DII, sulfur-diet score, HFD index, vitamin insufficiency, and mineral insufficiency), individual nutrient and food additive intakes, and food group consumptions—collectively constituting a holistic dietary profiling.

Participants' habitual dietary intake was evaluated using a validated food frequency questionnaire (FFQ) encompassing 250 food items, with consumption frequencies recorded over a three-month period. Nutrient intake was derived using the Food Processor Nutrition Analysis and Fitness Software (version 8.0, ESHA Research, Salem, USA). All dietary data were energy-adjusted to amounts per 1000 kcal of total daily energy, except for macronutrients, which were expressed as percentages of total energy (% en)[33]. Multiple indices were calculated to characterize diet quality and pro-inflammatory potential, with detailed compositional criteria provided in Supplementary Tables. Reflecting overall dietary balance, the CCDI (range: 0–140) is a measure specifically tailored to assess the dietary patterns of Chinese children, with higher scores indicating superior overall diet quality[34]. The dietary diversity, a core component of the CCDI, is strongly connected to the gut microbiome, reflecting its value in capturing diet-microbiome relationships[35]. DII was used as a validated measure of the pro-inflammatory potential of an individual's diet[36,37]. The HFD index, which was created by multiplying individual food consumptions by weighted health values and a Simpson's index score of all ingested items in accordance with the German Nutrition Society rules, is a useful tool for offering more insight into the variety of healthy diets[35,38]. The HFD index (range: 0–1) reflects dietary variety, with higher values denoting greater diversity. Adapted from the Harvard AHEI-2010[39], this index demonstrated robust associations with gut microbial composition in prior studies. Each component was scored against dietary recommendations, with elevated scores signifying improved diet quality[35]. Each component was scored against dietary recommendations, with elevated scores signifying improved diet quality[39]. Methodological details, including food transformation algorithms and scoring adjustments, are provided in the Supplementary Materials. Furthermore, micronutrient insufficiency status was quantified through vitamin insufficiency and mineral insufficiency, representing the cumulative count of vitamins and minerals, respectively, with intake levels below the Chinese Dietary Reference Intakes (DRIs) as defined by the national standard WS/T 578[40–42].

Exposures to food additive data were measured using a validated food additive questionnaire[43], which quantified annual intakes (in grams) and then normalized on a per-bodyweight basis (g/kg bw per year), as is the usual procedure in toxicology[44]. There were three types of emulsifiers/thickeners: polysorbate-80 [P80], carboxymethylcellulose [CMC], and carrageenan [CRN], with the sum reflecting total exposures as these components were commonly used in mixed-use[30].

### Phenotype data measurements

Systematic phenotypic assessment protocols were detailed in our prior study[8]. In brief, we implemented standardized instruments to evaluate core clinical domains, including the Chinese-validated Social Responsiveness Scale, 2nd edition (SRS-2)[45] and the 33-item Sensory Experiences Questionnaire (SEQ)[46], to evaluate the overall severity of ASD symptoms and sensory responsiveness. The Children's Eating Behavior Questionnaire (CEBQ) provided a multidimensional assessment of problems in responsiveness to food, less enjoyment, less satiety responsiveness, food fussiness, emotional overeating, emotional undereating, and desire for drinks[47]. Principal component analysis (PCA) was performed on CEBQ subscales to address the multidimensionality of eating behaviors (Supplementary Fig. 1B). Stool consistency was classified using the validated Bristol Stool Chart (7-type scale), serving as a proxy for bowel function[48]. Sociodemographic characteristics (parental education, employment status, household composition) were parent-reported. Continuous covariates were discretized into quartiles. Binary coding (yes/no) was applied for

comorbidities, GI symptoms, familial medical history, and medication usage.

### Fecal DNA sequencing and data analyses

To minimize potential batch effects during sample processing and sequencing, fecal samples were processed using a standard protocol[8]. Following preservative removal, microbial DNA was extracted using the Qiagen DNeasy PowerSoil Pro kit per the manufacturer's protocol. Post-quality assessment (Qubit 2.0, agarose gel electrophoresis, Agilent 2100), libraries were prepared with Illumina DNA Prep (M) Tagmentation through end repair, A-tailing, purification, and PCR amplification. Sequencing was performed on an Illumina NovaSeq system (150 bp paired-end). The ZymoBIOMICS Microbial Community (Standards D6300/D6306) served as positive controls throughout. Raw sequence data were quality filtered with Trimmomatic (v.39) to eliminate adapters, low-quality sequences (quality score < 20), and reads shorter than 50 base pairs. The remaining reads were then mapped to several mammalian genomes obtained from the UCSC Genome Browser, as well as bacterial plasmids and complete plastomes from the NCBI RefSeq database. Bowtie2 (v.2.4.2) was used for mapping, and potential host- and laboratory-associated contaminant reads were removed using KneadData v.0.6. To expedite data processing, GNU parallel (v.3.0) facilitated parallel analysis jobs. Taxonomic classification for metagenomic reads was performed using Kraken 2 (v.2.1.2) with an adoption of k-mer-based algorithms. To obtain precise estimates of taxonomic abundance, particularly at the

species and genus levels, Bracken (v.2.5.0) was utilized, building on the results from Kraken 2. The read counts for each species were then converted into relative abundances for subsequent analysis[49]. Furthermore, microbiome functional pathways were profiled using HUMAnN (v.3.0) and transformed into relative abundance before analysis.

Features with ultra-low prevalence (<10%) were originally filtered out[50,51]. Calculations of diversity richness (observed species and Shannon diversity) were performed using phyloseq (v1.24.2) and vegan (v2.6–4) in R. Considering that microbial data are sparse with a non-normal distribution, relevant statistics using relative abundance were performed using the ggpubr (v.0.6.0) package (https://github.com/kassambara/ggpubr). A dysbiosis score, calculated using Bray–Curtis dissimilarities to the gut microbiome compositions of the non-ASD group, was employed to assess the dysbiotic levels in autistic children[52]. Initially, a reference set was created from samples of non-ASD subjects. The dysbiosis score for each sample was defined as the median Bray–Curtis dissimilarity to this reference set, excluding samples from the same subject. To identify samples that significantly diverged from the reference, we set the threshold for the dysbiosis score at the 90th percentile of scores from non-ASD samples[52].

## Calculation of microbial features associated with diet and medications use

The correlations of diversity richness and gut dysbiosis score with dietary components within different groups were performed using Spearman's rank test, Kendall correlation method, and generalized regression model with adjustments of age, sex, body mass index, Bristol Stool Chart, GI conditions, and autistic symptoms. To address the issues of non-independence, centered-log-ratio transformation was conducted in the variance component and association analyses using the compositions (v2.0–5) R package. To evaluate the proportions of variance in microbiome composition or MetaCyc microbial functional pathways attributed to phenotypes, we applied permutational multivariate analysis of variance (PERMANOVA), derived from the adonis function in the vegan package of R (v.2.6–4). This approach focused on beta-diversity, employing the Bray–Curtis distance matrix generated from the relative abundances of microbial species, with 9999 permutations conducted. MaAsLin2 (v1.4.0, huttenhower.sph.harvard.edu/maaslin2) analysis was performed on microbial functionality and bacterial species linking to dietary factors after adjustments for age, sex, and GI conditions. The latter was selected over the Bristol Stool Chart due to its stronger association with overall microbial community variation. To evaluate whether diet-microbiome associations differed between groups, we fitted a regression model with an interaction term (microbe~diet + ASD + diet × ASD + age + sex + GI conditions). In addition to the default functional profiling using the HUMAnN database with MetaCyc summed pathway analysis, we further conducted KEGG Orthology (KO) analysis to enhance the mechanistic interpretation of microbial metabolic potential and functional capabilities. To ensure sufficient statistical power for within-group comparisons of ASD and non-ASD participants, we applied a 1:1 nearest-neighbor matching algorithm using the MatchIt (v3.7.2) package in R, matching on age, sex, and GI conditions.

Machine learning regression and binary classification models were employed to examine the relationships between actual dietary factors and microbiome-based dietary predictions, by generating correlation results and AUC values, respectively. AUC values were calculated using the pROC package (v1.18.5) in R. We used random forest classification and regression on both species-level taxonomic relative abundance and functional pathways (RF [v2.2] R package, ntree 1000, using default mtry), as the random forest models were demonstrated to have strong robustness and suitability in relation to microbiome abundance data[51]. For both the regression and

classification tasks, the Random Forest method was applied five times, with 10 training/testing folds was implemented.

## Microbial connection network and stability

The connection network among fecal bacterial species were calculated using the Sparcc method. The correlation matrix generated by the Sparcc package (v4.4.1). The topological features of the co-occurrence networks were generated by the iGraph package (v4.4.1) and visualized with node sizes proportional to relative abundances. To assess differences in microbial connection network, we compared ASD and non-ASD groups within tertiles of food emulsifier exposure, which were calculated across all participants to establish consistent risk strata. We assessed the significance of differences in topological features—such as reduced average node degree and clustering coefficient, which reflect ecological instability—using a non-parametric bootstrap procedure to compute bias-corrected p-values, with significance letters indicating $p < 0.05$ after multiple comparison adjustment[53]. Those nodes with the criteria: qval < 0.005 and |correlation coefficient| > 0.5 were colored based on the groups of modules, while the remaining nodes were in gray. The placement of the colored nodes was performed using the Fruchterman-Reingold layout, which utilizes a force-directed algorithm. We also applied the MDiNE model fitted by the Hamiltonian Monte Carlo method (1000 iterations) to estimate changes in the fecal microbial network topology within the groups[54].

## Statistical analysis

Statistical evaluations were conducted using R v.4.4.1. Participants were stratified into tertiles based on their CCDI scores. Between-group differences were assessed using Kruskal-Wallis tests for continuous variables and Pearson's Chi-squared tests for categorical variables, respectively. The tests were two-tailed, and exact p-values along with confidence intervals are reported. All the statistically significant diet-microbiome associations were further adjusted for multiple comparisons using false discovery rate (FDR) correction (Benjamini-Hochberg procedure). For each metadata panel, microbial species were ranked by the number of significant associations observed; then, the top 50 uniquely associated species per panel were retained for downstream analysis. In the association heatmaps, asterisks denote statistically significant associations (FDR-adjusted qval < 0.1).

## Ethics statement

This research complies with ethics regulations, with protocol approved by the Joint Chinese University of Hong Kong-New Territories East Cluster Clinical Research Ethics Committee (The Joint CUHK-NTEC CREC). Written consent was obtained from the children's parents.

## Reporting summary

Further information on research design is available in the Nature Portfolio Reporting Summary linked to this article.

# Data availability

The metagenomic sequencing data used in this study have been published before and were available in the NCBI Sequence Read Archive database under accession code PRJNA943687[8]. Processed dietary and microbial data are deposited in Zenodo (https://zenodo.org/) under the identifier No.17730818[55]. The full results of the diet-microbiome association analyses, including all exact p-values, are available in the Supplementary Data. The corresponding legends can be found in the Description of Additional Supplementary Files. Participant metadata cannot be made publicly available via repositories as outlined in the patient consent form to protect participant privacy. Requests for sharing metadata, including dietary profiles, can be submitted with a written proposal to the corresponding author (Prof. Siew C. Ng) at siewchienng@cuhk.edu.hk. The proposal should detail

the intended use of the data. The data management team, composed of scientists and clinicians, will review these requests based on scientific merit and ethical considerations, including patient consent, to avoid any misuse or misinterpretation. Data sharing will be undertaken if the proposed projects have a sound scientific rationale or potential patient benefit. Data recipients are required to enter a formal data sharing agreement, which describes the conditions for release and requirements for data transfer, storage, archiving, and publication. Since the data management meeting is held monthly, please anticipate a response within two working months. Data access is typically granted for 12 months under a Data Use Agreement that prohibits participant re-identification and third-party data transfer.

## Code availability
Scripts for data analysis steps in this manuscript are publicly available on Zenodo: https://zenodo.org/records/17730818[55].

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

## Acknowledgments

This study was supported by InnoHK (F.K.L.C., S.C.N.), the Government of Hong Kong, Special Administrative Region of the People's Republic of China, The D. H. Chen Foundation (F.K.L.C., S.C.N.), and the New Cornerstone Science Foundation through the New Cornerstone Investigator Program (S.C.N.).

## Author contributions

Y.Wu and O.W. conceived the study, ran analyses, and drafted the manuscript. S.Chen, Y.Wang, and W.L. contributed to part of the meta-genomic sequencing. C.P.C., J.Y.L.C., and P.K.C. contributed to subject recruitment, sample collection and biobank management. S.Chan and P.L. contributed to participant recruitment and clinical assessment. F.K.L.C. contributed to the study design and data interpretation. Q.S. and S.C.N. oversaw the entire study and contributed to the study design, data analysis and interpretation and manuscript writing. All authors gave final approval for the version to be published.

## Competing interests

F.K.L.C. serves as the Principal Investigator for the Fecal Microbiota Transplantation Service under the Hospital Authority (HA); a Board Direc-tor of EHealth Plus Digital Technology Ltd, an HA-owned subsidiary driving the eHealth+ program to transform the Electronic Health Record Sharing System into a comprehensive digital healthcare platform and advance other IT initiatives within the eHealth ecosystem; serves as a Director of the Hong Kong Investment Corporation Limited and a member of the Steering Committee for the RAISe+ Scheme under the Innovation and Technology Commissio; the Co-Director of the Microbiota I-Center (MagIC) Ltd. F.K.L.C. receives advisory fees and speaker honoraria from AstraZeneca and Comvita New Zealand Limited, as well as patent royalties through affiliated institutions for microbiome-related applications. F.K.L.C. and S.C.N. are shareholders of GenieBiome Holdings Limited and the co-founders, non-executive Board Chairman, and non-executive Scientific Advisor of its wholly owned subsidiary, G-NiiB GenieBiome Limited. F.K.L.C. is a shareholder of MicroSigX Diagnostic Holding Limited and the co-founder, non-executive Board Chairman, and non-executive Scientific Advisor of its wholly owned subsidiary, MicroSigX Biotech Diagnostic Limited. S.C.N. has served as an advisory board member for Pfizer, Ferring, Janssen and Abbvie and received honoraria as a speaker for Ferring, Til-lotts, Menarini, Janssen, Abbvie and Takeda; has received research grants through her affiliated institutions from Olympus, Ferring and Abbvie; is a founder member of MicroSigX Biotech Diagnostic Limited; is a share-holder of MicroSigX Diagnostic Holdings Limited; GenieBiome Limited is wholly owned by GenieBiome Holdings Limited; is a non-executive Board director and non-executive scientific advisor of MicroSigX Biotech Diag-nostic Limited and its holding company which is non-remunerative; and receives patent royalties through her affiliated institutions. QS, F.K.L.C., and S.C.N. are named inventors of patent applications held by the CUHK and MagIC that cover the therapeutic and diagnostic use of the micro-biome. The remaining authors declare no competing interests.
