## [Transparent Peer Review file · Nature Communications]

Distinct diet-microbiome associations in autism spectrum disorder

Corresponding Author: Professor Siew C. Ng

Version 0:

Reviewer comments:

Reviewer #1

(Remarks to the Author)

Wu et al. aimed to determine how the relationship between dietary intake and the gut microbiome differed between children (3-12 years-old) with and without autism spectrum disorder (ASD) through conducting extensive dietary surveys, phenotypic data measurements such as evaluation of ASD and GI symptom severity, and gut microbiome profiling via shotgun metagenomics sequencing of fecal samples. The size of the cohort (n=818) and breadth of data is impressive, and the research aim is both important and novel. However, the statistical analysis and presentation of the data in this current version of the manuscript hinders adequate interpretation of the data by the reader. We recommend major revisions.

Line [80]: "Diet-driven gut microbial signatures and phenotypic correlates in ASD"

In this section, it is stated that CCDI is significantly correlated with more severe autistic symptoms and GI complications. However, there are other diet metrics discussed in this manuscript that have ASD-specific diet-microbiome interactions: protein intake, bean consumption, P80, carrageenan, starch, and magnesium. Do these diet metrics correlate with severity of autistic symptoms or GI complications in the entire population, or specifically in the ASD population? Are the abundances of the microbial taxa or functions that were associated with these dietary metrics additionally correlated with severity of autistic symptoms or GI complications? Mediation and/or moderation analysis can be completed to determine if the microbiome mediates or moderates the association between diet and severity of autistic symptoms or GI complications.

In the current version of this manuscript, there are a lot of diet-microbiome interaction findings in ASD versus TD populations, but it is hard to determine if any of these findings are clinically relevant. It is important to demonstrate which dietary components, beyond just the dietary indices, exacerbate and also which dietary components alleviate the severity of autistic symptoms and GI complications both independently and by acting through the microbiome.

Line [108]: "While effect sizes for these factors were comparable in TD children, most associations lost statistical significance, underscoring the ASD-specific interplay between diet, phenotype, and gut ecology (Fig. 1D)."

There are two main issues with this statement. First, the sample size for the ASD population (n=462) is over 100 participants larger than the sample size for the TD population (n=356). Therefore, the loss in statistical significance could be due to the lower statistical power in the TD calculation compared to the ASD calculation. Second, the TD population could be more variable than the ASD population, because even though the TD population has an absence of neurological disorders, the TD population may not be completely "healthy" and may exhibit other morbidities relevant to microbiome dysbiosis such as gastrointestinal disorders. We recommend the following changes:

- 1) Effect sizes should be provided in the manuscript text for all important statistical findings. Results with similar effect sizes between groups should be interpreted with caution.
- 2) More phenotypic information should be provided on the TD population in order to understand its inherent variability.
 - a) Is there any overlap between the behavioral outcomes (SRS-2 and SEQ) in the ASD versus TD populations? If numbers permit, it may be best to remove patients in the TD population who were not diagnosed with ASD but had abnormal SRS-2 or SEQ scores to reduce population variability.
 - b) How many in the TD population versus the ASD population had irregular stools on the Bristol Stool Scale or had GI symptoms? If numbers permit, it would be interesting to do a sub-analysis on the ASD and TD populations without GI complications, and the ASD and TD populations with GI complications, to determine how ASD impacts diet-microbiome

interactions independently of GI complications, or how diet-microbiome interactions during GI complications are different in TD children and children with ASD.

3) As part of the limitations section, the difference in numbers between the TD and ASD populations and impact on statistics should be stated. Also, it should be mentioned that although the TD population was screened for neurological disorders, this population may contain children with other types of disorders. If you have comorbidity data, it would be relevant to state if any comorbidities were substantially present in the TD population.

[Line 127] "The breadth of diet-microbe associations varied by dietary variable: in ASD, protein intake (n=28), CCDI scores (n=27), and bean consumption (n=18) exerted the strongest effects on microbial species, whereas non-ASD children showed negligible linkages."

The total n for these analyses is quite low. Considering it was stated that 818 children were in the study, what is the reason for these low n? Was it a lack of completion of dietary surveys, a lack of fecal samples collected, or a lack of fecal samples sent out for shotgun metagenomics sequencing? An effort should be made to increase these numbers if possible. If not, interpretations from analyses with such low n should be made with caution, and the limitations of interpreting these analyses should be stated in the discussion section.

[Line 133]: "Moreover, predictive modeling using diet-associated species outperformed functional pathway-based predictions (Supplementary Fig. 3C), with random forest models demonstrating strong concordance between predicted and actual values for protein intake ($\rho > 0.35$) and CCDI scores."

These findings are unusual. One would expect a microbiome function to be tied to diet-microbiome interactions, which would yield important mechanistic clues. Two reasons on why these findings occurred that need to be addressed:

1. Due to the low n, these species level findings could be spurious, as there is much more variance in the presence/absence of individual species across individuals whereas the microbiome functions are much more conserved. For all findings involving microbial species, the number of individuals who had the species present in their gut microbiome should be stated.
2. The default HUMAnN database is the BioCyc/MetaCyc summed pathway abundances. The important microbiome functions may not be represented in this database. Better results may be yielded through the use of additional databases such as KEGG, GO, and CAZyme. It may also be better to conduct analyses on the individual orthologies rather than the summed pathways, and to subsequently complete gene-set enrichment analysis as a follow-up step.

Line [142]: "Given the prominence of microbial associations with CCDI scores and key nutrients (protein, starch, magnesium), we investigated gut microbial shifts in ASD children stratified by high intake of these dietary components." The microbial associations with the key nutrients (starch and magnesium) are not clearly stated in the previous sections, so this sentence is unclear to the reader.

Line [222] "Children with ASD exhibited pronounced ecological fragility in response to emulsifier exposure, marked by fragmented microbial networks and loss of keystone taxa."

Children with ASD also had lower microbiome alpha diversity than TD children. In this discussion section, we suggest that the authors link the findings of pronounced ecological fragility and fragmented microbial networks to the diminished alpha diversity, as it is known that lower diversity leads to higher interdependence between species-species relationships and species-substrate relationships within the microbial community.

Line [233]: "Limitations"

We think it is important to state as a limitation that the studied populations were relatively ethnically and culturally homogenous, and that as future work the study should be repeated in populations with different dietary patterns to increase diversity and validate the generalizability of this study's findings. Further, it should be stated if any of the dietary indices, dietary components, or food groups had low variability across the studied populations. For example, if the vast majority of the ASD population had low dark green vegetable intake or the TD population had high fish intake, this study would be limited in its ability to observe how the microbiome differs by dark green vegetable intake in the ASD population or fish intake in the TD population.

Reviewer #2

(Remarks to the Author)

The work does not align with established literature. For example, Zafar and Habib (2021) found lactobacillus and prevotella to be greatly decreased in the gastrointestinal tract of children with autism. The authors do not even look for these bacterial species in their study and yet, they claim to have analyzed "the top 50 microbial species."

The authors do not consider the impact of dietary heavy metals on gut microbiota when numerous studies indicate heavy metal exposures are involved in autism. Mangalam et al (2017) found a decline in Prevotella in response to heavy metal mixtures in the gut.

The study design has gaps in it which render the results of the study highly questionable in terms of their utility. The authors do not look at synthetic food additives which contain heavy metals (e.g., petroleum-based food colors). The methodology is not sound. I find this to be a poorly designed study because the authors clearly did not do a thorough literature review before they designed their study.

Reviewer #3

(Remarks to the Author)

Thank you for the opportunity to review Wu et al.'s manuscript.

The authors perform a metagenomics analysis of a large cohort accompanied by deep phenotyping information. As a starting point, they perform an important replication of previous work finding that poor dietary quality is associated with degree of autism symptoms and measures of microbiome diversity. In this well-powered dataset, they also find associations between poor dietary quality and gastrointestinal complications. The authors then go further, performing a variety of following analyses to characterise dietary-microbiome relationships.

A major strength of the paper is the depth of phenotyping information collected, particularly with respect to dietary measures, and how the authors have carefully explored potential confounders. I particularly like the data-driven analyses that have been performed in Section 2.2.

My questions largely relate to the logical flow of the paper which I believe could be improved to take the reader along on the journey. I often feel unsure what the "deep-dive" analyses (e.g., Section 2.3, 2.4) are being directed by, which can give the impression of cherry-picking (even if it is not). I also believe that the analyses need more statistical testing (rather than the largely qualitative presentation of results at current), plus technical and specific details to be included in the main text (examples below).

Major points:

- For Table 1, please justify why tertiles are used (and not quartiles, quintiles, etc.)? It would be better still if there were known cutoffs for CCDI (rather than being driven by the current data) that you could use to divide. Were tertiles used throughout the analysis (as is suggested in Methods line 403), or just for Figure 1? Please clarify – I believe that CCDI should be kept as a continuous variable in analyses or information is lost unnecessarily.
- Given there are various systematic differences shown in Table 1, please articulate what covariates have been included in each analysis, including in the main text.
- It is not immediately clear to me how the specific dietary measures shown in Fig.2A and Fig.2B were chosen. Fig. 2B in particular seems to be a subset of Fig.2A. How were these chosen? Were the dietary variables in Fig.2A also a subset and if so how were they chosen?
- For the Fig.2B analysis, is it possible that the difference in classification performance could be due to there being ~100 more people in the ASD group vs TD?
- In Section 2.3 Magnesium/protein enrichment and starch depletion in ASD – I am unclear in what way the protein, starch and magnesium associations are "prominent" (at least when going off Figure 2 where these variables do not stand out). Is this shown in a plot or table somewhere? If so, it would be helpful if this is referenced (and also should probably be in a main text figure). Without this clarity, I feel I am missing the relevance of this section.
- Similarly in Section 2.4, please point to what analysis led you to choose to focus on emulsifier exposures? Again, I think this should be in a main text figure.
- In Section 2.4, it would be help logical flow to prime the reader more to the relevance of network analyses and why you have used them.
- In Section 2.4, line 165, "dysbiosis" is mentioned. But what is the definition of dysbiosis here? Probably better to mention as ecological instability if some technical definition of dysbiosis is not met.
- In Section 2.4, please provide more specific metrics of the network differences with statistical testing if possible. At currently, the main text is entirely qualitative which I do not think is sufficient, and I believe Table 2 is unreferenced. Eg. I am not very familiar with network analyses (but the audience may also not be) but I am not immediately convinced that there is a difference between the ASD vs nonASD networks. Eg. "Neurotypical children, in contrast, maintained stable network architecture and functional resilience across emulsifier exposure levels". Please provide statistical tests to support such claims.
- In Section 2.4, how is a keystone species defined?
- For Figure 4 A, what is the x-axis label? For B/C, I'd suggest putting ASD and nonASD labels on the plots to make it easier to understand. What does "high exposures" mean here? For D/E, I would suggest including the nonASD results from SF6, if a key claim the authors are making is that ASD and nonASD networks are fundamentally different.
- For Section 2.5, I do not know what specific analyses have actually been performed. Could this please be clarified?
- In general, it would be helpful for the reader if the main text included some more details of methods than how the manuscript is currently written. Otherwise, the reader is required to flip back and forth between Methods and Results to make a critical assessment. Eg. For the analysis resulting in Fig.1D, could the authors please specify the model/analysis used in the main text rather than just the Methods (I believe this is PERMANOVA). For the Fig.2B analysis, could the authors please elaborate beyond "machine learning". Etc.

Minor points:

- Figure 2B could be improved by having the rectangle/circle described within the figure key (rather than just the legend).
- Is there a reason why carrageenan is abbreviated to CRN? It would make it easier to read if spelt out in full and wouldn't increase word count.
- There seems to be mention of both non-ASD and TD. Would suggest consistency unless this means something distinct.
- Reference 7 is cited as Johnson and Howell 2021 Cell Metabolism. However, this is commentary rather than the original primary analysis article which is Yap et al. 2021 Cell – I would suggest referencing the latter article (which to be transparent, I acknowledge is mine) as it was the first to empirically support dietary hypotheses of microbiome differences in autism.

Kind regards,
Chloe Yap

Reviewer #4

(Remarks to the Author)

Version 1:

Reviewer comments:

Reviewer #1

(Remarks to the Author)

The authors have done an excellent job with this revision. The findings and manuscript are much stronger having incorporated responses to all of the critiques. The results from their newly incorporated mediation analysis are particularly interesting.

Reviewer #3

(Remarks to the Author)

Thank you very much for the comprehensive replies and additional analyses completed by Wu et al.

My major remaining concern (apologies for only identifying one instance of this in my last review) relate to the validity of running separate analyses for ASD and nonASD groups, and then inferring differences based on a direct comparison of p-values between the groups. In essence this is a subgroup analysis of which an overview of pitfalls is provided here: <https://www.nejm.org/doi/10.1056/NEJMSr077003>. The p-value differences identified in these analyses may simply relate to differences in power between the groups, and the coefficients could just relate to different spread of data between groups. The correct analysis needed to identify whether there truly is a significantly different diet/microbiome relationship between the groups is to run a regression with an interaction term. Eg. $\text{microbe} \sim \text{diet} + \text{group} + \text{diet} * \text{group} + \text{covariates}$. If the $\text{diet} * \text{group}$ term is significant, then you can conclude that the groups have innately differently diet-microbiome associations. These analyses apply for all cases in which ASD and nonASD groups have been compared as subgroups.

Relatedly, I appreciate the inclusion of Supplementary Table 3, but for it to be interpretable, there needs to be a side-by-side comparison of 1) full-ASD group coefficients, 2) subset n=356 ASD group coefficients, 3) non-ASD group coefficients. It also needs to be expanded (probably as separate Supplementary Tables) as sensitivity analyses to stress-test other key results (eg. Firmicutes SGB4348, M funiformis, magnesium/protein-microbiome results, ideally the network results as well).

Specific comments

Paragraph starting line 94: could you please include coefficients and standard errors (currently just p-values)

Section 2.2: Apologies I missed this in the initial review – my (now) understanding of the analysis in this section is that the authors performed analyses within the ASD and nonASD groups separately, and then compare the q-value between the groups to identify “ASD-specific diet-microbiome interactions”. However, to me, a disparity in q-value seems more likely to reflect a difference in power between the groups: the ASD group is larger, and also has a greater spread of results (which would make the [coefficient] larger). Thus, I believe the correct analysis is to run the analysis with ASD+nonASD together, and including ASD/non-ASD status as an explanatory variable. Then I suggest reporting the coefficient for ASD/nonASD status, and identifying ASD-specific diet-microbiome interactions where the q-value for the ASD/nonASD term meets threshold (eg. line 134-146). I would expect these results to add more robustness to the idea that ASD and nonASD diet-microbiome associations are in some way distinct.

Line 124: It would be helpful to briefly justify why Bristol Stool Chart was not also included as a covariate

Line 139: “The decreased abundance of F.SGB4348 also indicated stronger correlations influenced by insufficient vitamin and mineral intake remarkably observed in ASD” – I’m not sure what “remarkably observed” means here

Reply to Figure 2B analysis classification performance: Thank you for this information. It would also help in the manuscript to state how you chose the matched subgroup.

Line 149, 167 – “exerted the most substantial effects”, “outsized role in modulating ...” implies causality which I don’t think can be determined from this analysis. Suggest rewording.

Line 179: Again, unclear whether the attenuated effect size could be a power issue / from there being less spread in the data. Again, suggest analysis with all individuals and extract coefficient for ASD/non-ASD status as an explanatory variable. Then for species associated with both the relevant dietary measures and ASD group status, can do GSEA analysis.

Paragraph starting line 190: please add statistics from mediation analysis.

Line 206: Please clarify whether children were split into tertiles when aggregated together or within the ASD/non-ASD

groups. I believe that the former is required to make a fair comparison.

Reply to Section 2.4 “dysbiosis” vs “ecological instability”: please also define exactly what is meant by ecological instability or whatever metric it is you use (apologies my comment before was unclear)

Reviewer #4

(Remarks to the Author)

Version 2:

Reviewer comments:

Reviewer #3

(Remarks to the Author)

Thank you very much to the authors for the additional analyses that have been performed. It is much appreciated.

Line 102-104: This analysis comparing effects between groups should include both ASD and non-autistic peers in a regression of: $CCDI \sim \text{Shannon diversity} + \text{ASDgroup} + \text{Shannon diversity} * \text{ASDgroup} + \text{covariates}$. You can then claim that the ASDgroup has a significantly stronger association if the interaction term is significant.

Sections 2.2 and 2.3 (relevant sections: lines 136-152; lines 184-193) Thank you for performing these analyses with the population-wide regression model + interaction term. The Supplementary Tables are very helpful and useful. However, I believe that the order of analyses should be the other way around. They are currently presented as 1) subgroup analysis 2) population analysis. However, I believe it should be 1) population analysis (ie. to test the question “Is there a significant ASDgroup difference in dietary feature X?”) 2) subgroup analysis to test the magnitude of the difference.

Paragraph starting line 203: please indicate whether the p-values reported are with multiple testing correction.

Other than these points (and updating the conclusions pending these results), I am satisfied with the manuscript.

Thank you again for the opportunity to review this work.

POINT-BY-POINT REPLY TO EDITORS AND REVIEWERS

Dear reviewers,

Thank you very much for the valuable comments to our manuscript. We have provided a point-by-point response to the reviewers' comments, together with a tracked and clean version of the revised manuscript. All changes are highlighted in yellow.

We appreciate the opportunity to resubmit and hope you find this version acceptable for publication.

Yours sincerely

Siew Ng on behalf of co-authors

Reviewers' Comments:

Reviewer #1:

Remarks to the Author:

Wu et al. aimed to determine how the relationship between dietary intake and the gut microbiome differed between children (3-12 years-old) with and without autism spectrum disorder (ASD) through conducting extensive dietary surveys, phenotypic data measurements such as evaluation of ASD and GI symptom severity, and gut microbiome profiling via shotgun metagenomics sequencing of fecal samples. The size of the cohort (n=818) and breadth of data is impressive, and the research aim is both important and novel. However, the statistical analysis and presentation of the data in this current version of the manuscript hinders adequate interpretation of the data by the reader. We recommend major revisions.

In this section, it is stated that CCDI is significantly correlated with more severe autistic symptoms and GI complications. However, there are other diet metrics discussed in this manuscript that have ASD-specific diet-microbiome interactions: protein intake, bean consumption, P80, carrageenan, starch, and magnesium. Do these diet metrics correlate with severity of autistic symptoms or GI complications in the entire population, or specifically in the ASD population? Are the abundances of the microbial taxa or functions that were associated with these dietary metrics additionally correlated with severity of autistic symptoms or GI complications? Mediation and/or moderation analysis can be completed to determine if the microbiome mediates or moderates the association between diet and severity of autistic symptoms or GI complications.

Response: We sincerely thank the reviewer for the positive comments. We categorized the Chinese Children Dietary Index (CCDI) rather than other dietary metrics to evaluate its association with the gut microbiota¹. We employed this approach because the CCDI reflects overall dietary quality and is specifically tailored to the dietary patterns and requirements of Chinese children, making it highly relevant to our study population. Furthermore, as supported by previous research, closer connections of healthy dietary index and diversity with the gut microbiota highlighted the importance of using a holistic dietary index—rather than focusing on isolated nutrients or single foods^{2,3}. We have revised the manuscript to clearly articulate the significance of the CCDI in the initial analysis (line 82-82, 381-385).

Line 82: Given the relevance of the Chinese Children Healthy Dietary Index (CCDI) for characterizing general dietary quality in our study population, we first conducted dietary profiling in a cohort of 818 children

Line 381: Reflecting overall dietary balance, the CCDI (range: 0 - 140) is a measure specifically tailored to assess the dietary patterns of Chinese children, with higher scores indicating superior overall diet quality³⁴. The dietary diversity, a core component of the CCDI, is strongly connected to the gut microbiome, reflecting its value in capturing diet-microbiome relationships.

We thank the reviewer for raising the important point regarding other diet metrics potentially correlated with autistic severity. We indeed found dietary intake and nutritional insufficiencies differ markedly between children with ASD and their non-ASD peers and correlate specifically with core symptoms within the ASD group. As shown in the following Figure, while consistent associations were observed between poor dietary quality (e.g., lower CCDI scores, pro-inflammatory diet) and hypersensitivity in ASD, specific nutrient-symptom associations often differed between ASD and non-ASD groups. Given the disease-specific dietary patterns and the potential for biased effect estimates in combined analyses, we analyzed the ASD and non-ASD groups separately instead of pooling these phenotypically distinct populations. This approach served as the methodological basis for the mediation analyses presented in the revised manuscript (line 191-196, Supplementary Fig. S7, Supplementary Table S8).

Figure Heatmap for the associations between diet and autistic phenotypes in children with/without ASD. The multivariable linear regression model was adjusted for age, sex, BMI, mother’s education, and GI—covariates previously identified to confound dietary intake.

To determine whether the diet-associated microbial species were additionally correlated with the severity of autistic symptoms, we conducted comprehensive mediation analyses within the ASD population (line 191-196) and included a discussion of their mechanistic implications

(line 281-286). GI conditions were treated as a confounder in the mediation analysis rather than as an outcome parallel to autistic symptoms. The complete results are available in Supplementary Table S8, with statistically significant mediation models visualized in Supplementary Fig. S7. Our analyses identified that intake of protein, zinc, and trans-fatty acids (TFA) was significantly linked to autistic symptoms, with these relationships being partially mediated by variations in gut microbial composition. Specifically, *Faecalibacterium prausnitzii* emerged as a potential beneficial mediator, attenuating the associations between dietary components and ASD symptomatology. Conversely, reduced abundances of *Firmicutes_SGB14874* and *Firmicutes_SGB71883*, observed in the context of higher TFA and lower zinc intake, were significantly associated with more severe ASD symptoms, highlighting their role in the adverse diet-microbiome-symptom pathway.

Line 191: To elucidate the role of microbiome in mediating these diet metrics associations in ASD, we performed mediation analyses. Among the specific nutritional components examined, *Faecalibacterium prausnitzii* exerted a beneficial mediating influence on the associations between protein/zinc intake and ASD symptomatology (Supplementary Fig. 7; Supplementary Table S8). Notably, *Coprococcus eutactus* dominated with the highest mediation proportion effects across all diet-ASD associations.

Line 281: Mediation analysis revealed that *Faecalibacterium prausnitzii* mediated the beneficial effects of protein and zinc intake on ASD symptoms, likely through its anti-inflammatory and butyrate-producing properties²⁶. *Coprococcus eutactus* exhibited the strongest mediation effect, suggesting a key role in gut-brain communication and potential as a probiotic target for dietary interventions^{27,28}. These findings highlight the possibility of microbiota-directed therapies for ASD.

- [1] Duan, R., Qiao, T., Chen, Y. et al. The overall diet quality in childhood is prospectively associated with the timing of puberty. *Eur J Nutr.* 2021. 60, 2423–2434. Doi: 10.1007/s00394-020-02425-8
- [2] Asnicar F, Berry SE, Valdes AM, et al. Microbiome connections with host metabolism and habitual diet from 1,098 deeply phenotyped individuals. *Nat Med.* 2021, 27(2):321-332. doi: 10.1038/s41591-020-01183-8.
- [3] Bowyer, R.C.E., Jackson, M.A., Pallister, T. et al. Use of dietary indices to control for diet in human gut microbiota studies. *Microbiome.* 2018. 6, 77. doi: 10.1186/s40168-018-0455-y

In the current version of this manuscript, there are a lot of diet-microbiome interaction findings in ASD versus TD populations, but it is hard to determine if any of these findings are clinically relevant. It is important to demonstrate which dietary components, beyond just the dietary indices, exacerbate and also which dietary components alleviate the severity of autistic symptoms and GI complications both independently and by acting through the microbiome.

Response: Thank you for this insightful comment. In response, we have now conducted comprehensive mediation analyses, adjusting for age, sex, and GI conditions, to identify specific microbial taxa that significantly mediate the relationships between dietary factors and ASD severity (Supplementary Fig. S7 and Table S8). This analysis screened all diet-associated microbial features to prioritize those with the greatest potential clinical relevance as mediators.

The results revealed several promising candidates; for instance, *Faecalibacterium prausnitzii* appeared to modulate the impact of diet on symptom severity, suggesting their potential as therapeutic targets. We have expanded the Discussion (line 191-196, 281-286) to incorporate these findings and emphasize their clinical implications.

Line [108]: “While effect sizes for these factors were comparable in TD children, most associations lost statistical significance, underscoring the ASD-specific interplay between diet, phenotype, and gut ecology (Fig. 1D).”

There are two main issues with this statement. First, the sample size for the ASD population (n=462) is over 100 participants larger than the sample size for the TD population (n=356). Therefore, the loss in statistical significance could be due to the lower statistical power in the TD calculation compared to the ASD calculation. Second, the TD population could be more variable than the ASD population, because even though the TD population has an absence of neurological disorders, the TD population may not be completely “healthy” and may exhibit other morbidities relevant to microbiome dysbiosis such as gastrointestinal disorders. We recommend the following changes:

1) Effect sizes should be provided in the manuscript text for all important statistical findings. Results with similar effect sizes between groups should be interpreted with caution.

Response: Thank you for your feedback regarding the comparison of effect sizes between the ASD and non-ASD populations. We have supplemented the manuscript with a formal comparison of effect sizes and species prevalences across all groups to ensure transparent and clear presentation (Supplementary Table S4). We acknowledge the important points that statistical power depends on sample size and that we cautiously compared effect sizes between the ASD and non-ASD groups. As highlighted in the revision (line 133-142), typical dietary metrics—CCDI, DII, and vitamin/mineral insufficiency—demonstrated stronger effect sizes on *Firmicutes SGB4348* and *Megamonas funiformis*, along with significant q-values in ASD. We also included the sensitivity analysis using a 1:1-matched ASD subgroup (n = 356) with an equal number of non-ASD participants (Supplementary Table S3, line 104-106). Notably, even with this reduced sample size, CCDI exhibited more significant associations with microbial features in the ASD group, reinforcing the validity and clinical relevance of our initial results.

Line 133: Specifically, *Firmicutes SGB4348* and *Megamonas funiformis* emerged as indicators of healthier dietary patterns in ASD, characterized by higher CCDI scores (*F.SGB4348*: coef = 0.64, qval = 0.01 in ASD vs. coef = 0.18, qval = 0.79 in non-ASD; *M.funiformis*: coef = 0.56, qval = 0.01 in ASD vs. coef = 0.24, qval = 0.91 in non-ASD) and lower DII scores (*F.SGB4348*: coef = -0.49, qval=0.09 in ASD vs. coef = -0.32, qval = 0.84 in non-ASD; *M.funiformis*: coef = -0.41, qval=0.10 in ASD vs. coef = -0.38, qval = 0.51 in non-ASD). The decreased abundance of *F.SGB4348* also indicated stronger correlations influenced by insufficient vitamin and mineral intake remarkably observed in ASD ($|\text{coef}| > 0.5$ and $\text{qval} < 0.1$, Supplementary Table S4).

2) More phenotypic information should be provided on the TD population in order to understand its inherent variability.

a) Is there any overlap between the behavioral outcomes (SRS-2 and SEQ) in the ASD versus TD populations? If numbers permit, it may be best to remove patients in the TD population

who were not diagnosed with ASD but had abnormal SRS-2 or SEQ scores to reduce population variability.

Response: We appreciate the Reviewer’s insightful suggestion regarding the importance of reducing heterogeneity within the non-ASD by accounting for abnormal behaviors. We would like to clarify, however, that the non-ASD participants in our cohort were carefully characterized to minimize heterogeneity: all scored within the normal ranges on the Social Responsiveness Scale (SRS) and the Sensory Experience Questionnaire (SEQ), and were free of neurodevelopmental and psychiatric comorbidities, as detailed in the demographic table below. This suggests a relatively homogeneous comparison group suitable for identifying ASD-specific associations. Furthermore, to address the potential confounding effect of GI symptoms—which were already adjusted for in our primary models—we added a sensitivity analysis by excluding the small subset of non-ASD individuals with GI complications (line 258-263, Supplementary Fig. S10). Please attend the following characteristic table for reference. We have supplemented this information in the main text to elaborate on the potential bias due to variability.

Line 307: The absence of neurodevelopmental and psychiatric comorbidities in the non-ASD group minimized the heterogeneity of the population and solidified our non-significant findings (Supplementary Table S9).

Line 310: Despite data variability of diet was sufficient for the primary analyses, the limited range in certain micronutrients (e.g., B vitamins) may have reduced sensitivity for detecting subtler associations (Supplementary Table S9).

Supplementary Table S9:

	ASD N=462	Non-ASD N=356	p value
Age	8.00 (6.00-10.0)	9.00 (8.00-10.0)	<0.001
Male	395 (85.5%)	200 (56.2%)	<0.001
BMI	16.0 (14.6-18.1)	16.3 (14.8-18.7)	0.056
Family medical history	93 (20.1%)	19 (5.34%)	<0.001
Comorbidity (yes)	259 (56.1%)	0 (0.00%)	<0.001
Medication	150 (32.5%)	0 (0.00%)	<0.001
GI symptoms	92 (19.9%)	19 (5.34%)	<0.001
Bristol stool chart			0.050
type 1	25 (5.41%)	11 (3.09%)	
type 2	67 (14.5%)	45 (12.6%)	
type 3	100 (21.6%)	84 (23.6%)	
type 4	160 (34.6%)	146 (41.0%)	
type 5	29 (6.28%)	31 (8.71%)	
type 6	9 (1.95%)	1 (0.28%)	
type 7	1 (0.22%)	0 (0.00%)	
Autistic symptoms			
SRS_RRB	62.0 (55.0-71.0)	46.0 (43.0-50.0)	<0.001
SRS_SCI	68.0 (61.0-75.0)	53.0 (49.0-58.0)	<0.001
SEQ_hyper	2.07 (1.79-2.50)	1.57 (1.36-1.86)	<0.001

SEQ_hypo	1.67 (1.33-2.17)	1.33 (1.00-1.50)	<0.001
Eating behaviors (CEBQ)	0.17 (-0.42-0.79)	-0.26 (-0.82-0.37)	<0.001

b) How many in the TD population versus the ASD population had irregular stools on the Bristol Stool Scale or had GI symptoms? If numbers permit, it would be interesting to do a sub-analysis on the ASD and TD populations without GI complications, and the ASD and TD populations with GI complications, to determine how ASD impacts diet-microbiome interactions independently of GI complications, or how diet-microbiome interactions during GI complications are different in TD children and children with ASD.

Response: We thank the reviewer for this insightful suggestion. We agree that this is a crucial question for disentangling the specific effects of ASD from those of co-occurring GI conditions. We therefore adjusted for GI conditions in all original association analyses in the main text. We have conducted additional sub-analyses stratifying both the ASD and non-ASD populations by the presence of GI complications and have incorporated a discussion of these results and their implications into the revised manuscript (line 263–268 and 314–316). To ensure statistical power given the low number of non-ASD children with GI symptoms (n = 19), we defined GI complications broadly to include both parent-reported GI symptoms and irregular stool types (Bristol Stool Scale types other than 3 or 4). The results, visualized in Supplementary Fig. S10, revealed that the most significant and consistent diet-microbiome interactions were observed in ASD children without GI complications. This pattern suggests that GI comorbidities may alter or obscure microbial responses to dietary intake, highlighting the importance of considering GI status when designing microbiome-targeted dietary interventions for children with ASD.

Line 263: This critical question—regarding how heterogeneity in GI conditions modulates diet-microbiome interactions—prompted a focused sub-analysis within the ASD population, stratified by the presence of GI symptoms and irregular Bristol Stool types. It revealed that dietary indices exerted a more pronounced influence on microbial features in ASD children without GI complications (Supplementary Fig. 10), suggest that GI health appears to be a prerequisite for robust diet-microbiome interactions 20.

Line 314: Meanwhile, our findings positioned GI health as a potential effect modifier of diet-microbiome interactions, necessitating careful consideration in subgroups to avoid biased interpretations.

3) As part of the limitations section, the difference in numbers between the TD and ASD populations and impact on statistics should be stated. Also, it should be mentioned that although the TD population was screened for neurological disorders, this population may contain children with other types of disorders. If you have comorbidity data, it would be relevant to state if any comorbidities were substantially present in the TD population.

Response: We appreciate your suggestion to include this clarification in the limitations section. Regarding comorbidity data, we have reviewed our study's metadata, which indicates that there are no substantial comorbidities present in the non-ASD children, as shown in the supplementary table S9 in response to your previous comment. This information was clearly articulated in the revised manuscript (line 303-309).

Line 303: However, limitations warrant caution: (i) The cross-sectional design precludes causal inference, and the discrepant sample sizes between ASD and non-ASD groups may influence statistics. Nevertheless, sensitivity analyses using matched subgroup sizes consistently confirmed the distinct diet-microbiome associations observed in ASD. The absence of neurodevelopmental and psychiatric comorbidities in the non-ASD group minimized the heterogeneity of the population and solidified our non-significant findings (Supplementary Table S9).

[Line 127] “The breadth of diet-microbe associations varied by dietary variable: in ASD, protein intake (n=28), CCDI scores (n=27), and bean consumption (n=18) exerted the strongest effects on microbial species, whereas non-ASD children showed negligible linkages.”

The total n for these analyses is quite low. Considering it was stated that 818 children were in the study, what is the reason for these low n? Was it a lack of completion of dietary surveys, a lack of fecal samples collected, or a lack of fecal samples sent out for shotgun metagenomics sequencing? An effort should be made to increase these numbers if possible. If not, interpretations from analyses with such low n should be made with caution, and the limitations of interpreting these analyses should be stated in the discussion section.

Response: We thank the reviewer for this valuable feedback and apologize for any misunderstanding caused by our initial description. We would like to clarify that all association analyses utilized the full sample size. The values n = 28, n = 27, and n = 18 refer

to the number of microbial species found to be significantly associated with dietary components—not the sample size of the study population. We have revised the relevant section in the Results (line 148–156) to prevent potential misinterpretation.

Line 148: The breadth of diet-microbe associations varied by dietary variable, showing stronger connections in ASD compared to their non-ASD peers, who exhibited negligible linkages. The dietary factors that exerted the most substantial effects on microbial species—across nutritional components, dietary indices, and food groups—were protein intake (28 significant associations in total), CCDI scores (27 in total), and bean consumption (18 in total) (Fig. 2B; Supplementary Fig. 4B). By looking into specific nutrients in relation to microbial shifts, starch and magnesium emerged as the nutrients with the second-highest number of significant associations after protein intake, exhibiting 27 and 20 significant associations, respectively (Fig. 2B).

[Line 133]: “Moreover, predictive modeling using diet-associated species outperformed functional pathway-based predictions (Supplementary Fig. 3C), with random forest models demonstrating strong concordance between predicted and actual values for protein intake ($\rho > 0.35$) and CCDI scores.”

These findings are unusual. One would expect a microbiome function to be tied to diet-microbiome interactions, which would yield important mechanistic clues. Two reasons on why these findings occurred that need to be addressed:

1. Due to the low n , these species level findings could be spurious, as there is much more variance in the presence/absence of individual species across individuals whereas the microbiome functions are much more conserved. For all findings involving microbial species, the number of individuals who had the species present in their gut microbiome should be stated.

Response: We thank the reviewer for raising this important point. To clarify, all correlation analyses between diet and microbial species were conducted using the complete dataset, wherein full dietary information was available for all participants, as stipulated in the Methods. No samples were excluded due to missing dietary metrics. In direct response to this comment, we have now included Supplementary Table S4, which specifies the number of individuals in which each bacterial species was detected. We also confirm that downstream association analyses were restricted to microbial species with a detection prevalence exceeding 10% of samples (line 443), thereby reducing the risk of spurious associations arising from low-prevalence taxa. Furthermore, we have revised the wording in the relevant Results section to more explicitly frame these findings as exploratory and to highlight the necessity for validation in larger, independent cohorts. Thank you for helping improve the clarity and rigor of our manuscript.

2. The default HUMAnN database is the BioCyc/MetaCyc summed pathway abundances. The important microbiome functions may not be represented in this database. Better results may be yielded through the use of additional databases such as KEGG, GO, and CAZyme. It may also be better to conduct analyses on the individual orthologies rather than the summed pathways, and to subsequently complete gene-set enrichment analysis as a follow-up step.

Response: We appreciate your insight regarding the limitations of the HUMAnN database. In

response, we have included KEGG orthology (KO) analyses to enhance the mechanistic understanding of the findings. Among the dietary metrics, healthy dietary indices, especially CCDI score, exhibited strong associations with predicted gene functions as the heatmap shown (Supplementary Fig. S3). The subsequent gene-set enrichment analysis further demonstrated significant contributions of amino-acids, particularly Phenylalanine, tyrosine and tryptophan biosynthesis, involved in the diet-microbiome interaction. We have included detailed descriptions in the main text (line 184-188, 222-224, Supplementary Fig. S6E, Fig. S8E).

Line 184: Concurrently, gene-set enrichment analysis based on KEGG orthology (KO) indicated enhanced microbial processing functions related to amino acid and cofactor biosynthesis (Supplementary Fig. 6E), suggesting a state of metabolic dysregulation in children with ASD linked to dietary imbalances.

Line 222: KEGG enrichment analysis further revealed that polysorbate-80/carrageenan exposure concurrently downregulated the 2-oxocarboxylic acid metabolism and citrate cycle pathways (Supplementary Fig. 8E).

Line 274: Complementary KEGG enrichment analysis linked CCDI-associated orthologs to cofactor biosynthesis (e.g., thiamine metabolism), suggesting its role in the improved ASD-related outcomes²³.

Line [142]: “Given the prominence of microbial associations with CCDI scores and key nutrients (protein, starch, magnesium), we investigated gut microbial shifts in ASD children stratified by high intake of these dietary components.”

The microbial associations with the key nutrients (starch and magnesium) are not clearly stated in the previous sections, so this sentence is unclear to the reader.

Response: Thank you for your valuable comment. We apologize for the oversight regarding the logic flow, as the key nutrients were not clearly stated in the previous section. The associations of these nutrients (starch and magnesium) with microbial shifts were identified as significantly related to the microbiome, exhibiting the second-highest numbers of significant associations after protein intake. We have revised the wording in the manuscript to clarify these connections for the reader (line 150-156).

Line 150: The dietary factors that exerted the most substantial effects on microbial species—across nutritional components, dietary indices, and food groups—were protein intake (28 significant associations in total), CCDI scores (27 in total), and bean consumption (18 in total) (Fig. 2B; Supplementary Fig. 4B). By looking into specific nutrients in relation to microbial shifts, starch and magnesium emerged as the nutrients with the second-highest number of significant associations after protein intake, exhibiting 27 and 20 significant associations, respectively (Fig. 2B).

Line [222] “Children with ASD exhibited pronounced ecological fragility in response to emulsifier exposure, marked by fragmented microbial networks and loss of keystone taxa.” Children with ASD also had lower microbiome alpha diversity than TD children. In this discussion section, we suggest that the authors link the findings of pronounced ecological fragility and fragmented microbial networks to the diminished alpha diversity, as it is known that lower diversity leads to higher interdependence between species-species relationships and

species-substrate relationships within the microbial community.

Response: We thank the reviewer for this constructive suggestion. We agree that the fragmented microbial networks in children with ASD are closely linked to their reduced alpha diversity. As illustrated in supplementary Figure S1C, higher exposure to P80 and CRN appeared strongly correlated with decreased alpha diversity and elevated dysbiosis index—a relationship that was especially observed in the ASD group. We have incorporated this perspective into the Discussion section (line 290–293) to more thoroughly contextualize how dietary emulsifier exposure may contribute to diversity loss and exacerbate microbial network instability in ASD.

Line 290: The significant loss of alpha diversity and the exacerbated dysbiosis associated with increased exposure to carrageenan/polysorbate-80 in autistic children (Supplementary Fig. 1C) offered a plausible mechanism for the pronounced network instability observed specifically in ASD following the dietary disturbance.

Line [233]: “Limitations”

We think it is important to state as a limitation that the studied populations were relatively ethnically and culturally homogenous, and that as future work the study should be repeated in populations with different dietary patterns to increase diversity and validate the generalizability of this study’s findings. Further, it should be stated if any of the dietary indices, dietary components, or food groups had low variability across the studied populations. For example, if the vast majority of the ASD population had low dark green vegetable intake or the TD population had high fish intake, this study would be limited in its ability to observe how the microbiome differs by dark green vegetable intake in the ASD population or fish intake in the TD population.

Response: We sincerely thank the reviewers for this constructive comment. We have now extensively revised the Discussion section to emphasize that the relatively ethnically and culturally homogenous populations may limit the generalizability of our findings. We have included a call for future studies to validate our results in more diverse cohorts (line 313-314). We also acknowledged concerns regarding dietary metrics with low variability. A review of our metadata (included in Supplementary Table S9) indicates that variability is similar across both ASD and non-ASD populations. We have added a sentence stating that while variability was sufficient to detect the reported associations, the restricted range in some micronutrients, such as vitamin B groups, might have limited our power to detect more subtle relationships involving them (line 310-312).

Line 310: Despite data variability of diet was sufficient for the primary analyses, the limited range in certain micronutrients (e.g., B vitamins) may have reduced sensitivity for detecting subtler associations (Supplementary Table S9). As dietary habits are culturally embedded, our findings require validation in more diverse populations to ensure generalizability.

Line 313: As dietary habits are culturally embedded, our findings require validation in more diverse populations to ensure generalizability.

Part of the Supplementary Table S9:

Characteristics	ASD n=462	Non-ASD n=356	p value
Dietary variables			

AHEI	53.9 (20.6-81.4)	54.7 (26.8-86.3)	0.059
CCDI	74.3 (40.5-111)	79.0 (34.5-116)	<0.001
DII	0.295 (-4.09-4.19)	-0.220 (-4.27-4.30)	0.022
Sulfurdiet score	0.28 (-4.85-6.47)	-0.06 (-4.17-3.96)	<0.001
HFD	0.46 (0.20-0.69)	0.48 (0.26-0.76)	0.003
Mineral insufficiency	5.00 (1.00-12.0)	4.00 (0-12.0)	0.698
Vitamin insufficiency	7.00 (1.00-10.0)	7.00 (1.00-10.0)	0.861
Polysorbate-80	7.91 (0.36-40.9)	6.38 (0.20-33.9)	<0.001
Carboxymethylcellulose	16.2 (0-90.6)	12.2 (0.08-55.0)	<0.001
Carrageenan	9.96 (0.28-57.6)	7.85 (0.48-40.6)	<0.001
Emulsifiers	35.1 (1.28-179)	27.4 (1.28-95.4)	<0.001
Protein (% of total energy)	14.9 (7.82-27.5)	16.5 (9.92-27.5)	<0.001
Fat (% of total energy)	30.6 (18.0-52.7)	29.7 (16.6-45.4)	0.600
Carbohydrate (% of total energy)	54.3 (17.9-73.9)	54.2 (29.4-72.1)	0.652
SFA (% of total energy)	10.1 (4.30-16.4)	10.3 (4.41-19.2)	<0.001
TFA (% of total energy)	0.17 (0.01-1.93)	0.15 (0-1.21)	0.181
PUFA (% of total energy)	4.77 (1.57-11.5)	4.08 (1.55-8.84)	<0.001
MUFA (% of total energy)	7.72 (0.61-21.8)	7.05 (0.767-15.1)	0.007
n3/n6	0.12 (0.04-0.42)	0.13 (0.07-0.29)	0.136
Monosaccharide (g/1000 kcal)	6.71 (0-55.8)	7.96 (0-35.2)	0.016
Disaccharide (g/1000 kcal)	3.77 (0-26.9)	4.91 (0-24.5)	0.004
Starch (g/1000 kcal)	76.7 (15.8-136)	74.2 (32.0-139)	0.033
Fiber (g/1000 kcal)	0.78 (0.09-11.30)	0.89 (0.19-4.51)	0.002
Added Sugar (g/1000kcal)	2.15 (0-50.9)	3.67 (0-27.8)	<0.001
Calcium (mg/1000 kcal)	267 (96.9-2170)	288 (78.8-972)	0.182
Phosphorus (mg/1000 kcal)	463 (189-1170)	495 (241-840)	<0.001
Sodium (mg/1000 kcal)	1190 (308-2700)	1290 (421-2770)	0.009
Potassium (mg/1000 kcal)	980 (464-2210)	1050 (507-2990)	<0.001
Magnesium (mg/1000 kcal)	102 (41.8-184)	107 (40.2-236)	0.001
Chromium (mg/1000 kcal)	2.69 (0.618-14.0)	2.69 (0.417-10.5)	0.884
Copper (mg/1000 kcal)	0.433 (0.15-1.53)	0.467 (0.18-1.49)	<0.001
Iodine (mcg/1000 kcal)	33.6 (0.28-195)	35.5 (1.54-193)	0.073
Iron (mg/1000 kcal)	5.10 (1.96-26.4)	5.45 (2.05-17.4)	0.048
Molybdenum (mcg/1000 kcal)	6.72 (0-33.1)	7.69 (0-47.8)	0.002
Selenium (mcg/1000 kcal)	44.5 (13.4-274)	48.7 (19.6-112)	0.014
Zinc (mg/1000 kcal)	4.10 (1.61-12.1)	4.34 (2.04-11.6)	0.020
Vitamin B1 (mg/1000 kcal)	0.43 (0.145-1.56)	0.437 (0.12-1.23)	0.993
Vitamin B2 (mg/1000 kcal)	0.57 (0.16-3.21)	0.57 (0.12-1.68)	0.198
Vitamin B3 (mg/1000 kcal)	6.32 (1.34-17.9)	6.68 (1.47-14.1)	0.014
Pantothenic acid (mg/1000 kcal)	2.01 (0.72-8.36)	2.10 (0.61-4.42)	0.045
Vitamin B6 (mg/1000 kcal)	0.31 (0.06-1.24)	0.30 (0.08-1.11)	0.614
Vitamin B12 (mg/1000 kcal)	0.292 (0-2.96)	0.328 (0-4.08)	0.662
Biotin (mcg/1000 kcal)	9.21 (1.26-57.9)	10.2 (1.22-43.3)	0.003
Vitamin C (mg/1000 kcal)	40.3 (1.65-254)	43.3 (0.94-235)	0.088

Vitamin D (mcg/1000 kcal)	1.62 (0.12-11.4)	1.77 (0.04-8.25)	0.423
Folate (mcg/1000 kcal)	89.1 (17.4-323)	96.2 (11.2-297)	<0.001
Vitamin K (mcg/1000 kcal)	29.5 (2.32-342)	34.3 (0.01-350)	0.042
Caffeine (mg/1000 kcal)	0.0262 (0-11.8)	0.956 (0-23.4)	<0.001
Chlorophyll (mg/1000 kcal)	128 (25.8-453)	141 (42.0-747)	<0.001
Fruit (g/1000 kcal)	113 (0-710)	145 (0-1490)	<0.001
Vegetable (g/1000 kcal)	145 (0-1050)	194 (0-1830)	<0.001
Sugarjam (g/1000 kcal)	4.65 (0-175)	4.29 (0-77.1)	0.105
Processed meat (g/1000 kcal)	18.8 (0-266)	18.0 (0-163)	0.265
Fried food (g/1000 kcal)	37.3 (0-180)	40.4 (0-178)	0.741
Dark green vegetables (g/1000 kcal)	20.0 (0-166)	28.6 (0-480)	<0.001
Dairy (g/1000 kcal)	66.5 (0-1000)	81.1 (0-773)	0.420
Beans (g/1000 kcal)	15.4 (0-271)	22.1 (0-196)	0.015

Macronutrients were presented as percentages of total energy intake. SRS = the Social Responsiveness Scale; SEQ = the Sensory experiences questionnaire; CCDI = the Chinese Children Healthy Dietary Index; AFEI = Alternative Healthy Eating Index; DII = Dietary Inflammatory Index; HFD = healthy food diversity index; SFA = saturated fatty acids; MUFA = monounsaturated fatty acids; PUFA = polyunsaturated fatty acids; TFA = trans-fatty acids.

Reviewer #2 (Remarks to the Author):

The work does not align with established literature. For example, Zafar and Habib (2021) found lactobacillus and prevotella to be greatly decreased in the gastrointestinal tract of children with autism. The authors do not even look for these bacterial species in their study and yet, they claim to have analyzed "the top 50 microbial species."

Response: We thank the reviewer for highlighting this important point regarding alignment with the existing literature. We appreciate the opportunity to clarify the focus and methodology of our study. Our work specifically investigated the diet-microbiome interaction in the context of autism, rather than focusing solely on case-control taxonomic differences. We identified "the top 50 microbial species" that, while not classified as ASD-specific according to case-control analyses, demonstrated significant interactions with dietary factors. The primary aim was to identify how dietary factors shape the gut microbiota in children with ASD and whether these relationships differ from those in neurotypical children. Although well-characterized taxa such as *Lactobacillus* and *Prevotella*, which are depleted in ASD, are certainly of interest, their responses to dietary factors were not prominently observed in our study, limiting their relevance for interpreting the diet-microbiome interaction.

The authors do not consider the impact of dietary heavy metals on gut microbiota when numerous studies indicate heavy metal exposures are involved in autism. Mangalam et al (2017) found a decline in *Prevotella* in response to heavy metal mixtures in the gut. The study design has gaps in it which render the results of the study highly questionable in terms of their utility. The authors do not look at synthetic food additives which contain heavy metals (e.g., petroleum-based food colors). The methodology is not sound. I find this to be a poorly designed study because the authors clearly did not do a thorough literature review before they designed their study.

Response: We thank the reviewer for raising this critical point. We appreciate the reviewer's comments regarding the impact of dietary heavy metals on gut microbiota and the potential gaps in our study design. While we acknowledge the importance of heavy metals, their presence as contaminants in foods is not a direct dietary component under voluntary consumption. Our aim is to identify modifiable dietary components that directly affect the gut microbiome—with the ultimate goal of informing microbiome-targeted dietary interventions. We would like to claim very low risk of heavy metal exposure obtained from food additives, since the validated assessment tool on nine types of food additives commonly consumed in Hong Kong^{1,2} demonstrating artificial stabilizers should be the most serious problem less likely contained heavy metal. Recent literature also highlights the elevated risks associated with emulsifiers in Hong Kong's children, demonstrating a closer relationship with neurodevelopmental disorders and gut microbiota³. This supports our decision to concentrate on these stabilizers. Moreover, it is reported that in highly developed regions such as Hong Kong^{4,5}, heavy metal exposure would be generally well-regulated and relatively low compared with other regions. Our own ancillary analysis of other additives, including titanium dioxide (which may contain trace metals), revealed no significant between-group differences or substantial variation in exposure levels. This further supports the notion that heavy metals are unlikely to be the primary driver in our cohort.

Lastly, we fully agree that environmental pollutants including heavy metals warrant further investigation. We have now explicitly addressed this point in the Limitations and Future Directions section (line 316-321), emphasizing the need for future studies to integrate multi-pollutant models and expand contaminant monitoring in dietary ASD research.

Line 316: Emulsifier effects may be compounded by unmeasured additives, and dietary heavy metals, as a risk factor, may also interact with the gut microbiome. Future studies specifically designed to incorporate biomarker-based assessments or utilize detailed dietary databases with comprehensive additive composition are urgently needed to elucidate the role of these substances.

[1] Trakman GL, Lin W, Wilson-O'Brien AL, Stanley A, Hamilton AL, Tang W, Or L, Ching J, Morrison M, Yu J, Ng SC, Kamm MA. Development and Validation of Surveys to Estimate Food Additive Intake. *Nutrients*. 2020;12(3):812. doi: 10.3390/nu12030812.

[2] Loayza JJJ, Kang S, Schooth L, et al. Effect of food additives on key bacterial taxa and the mucosa-associated microbiota in Crohn's disease. The ENIGMA study. *Gut Microbes*. 2023;15(1):2172670. doi: 10.1080/19490976.2023.2172670.

[3] Liu Y, Wang S, Xu W, et al. The impact of food additive intake and altered gut microbiome on perinatal health: data from 3 regions in China from the MOMMY COHORT. *Gastroenterology*. 2024; 166 (5):S-272 - S-273. Doi: 10.1016/S0016-5085(24)01111-9.

[4] Hou, W., Hu, R., Cui, Z. et al. Heavy metals in urban and rural indoor dust in plateau homes: levels, sources, and health risks. *Aerosol Sci Eng* (2025). <https://doi.org/10.1007/s41810-025-00308-1>

[5] Luo L, Wang L, Li Y, Cao H, Guo Y, Liao X. Urban-rural inequality in soil heavy metal health risks: Insights from Baoding, China. *Ecotoxicol Environ Saf*. 2025; 300:118458. doi: 10.1016/j.ecoenv.2025.118458.

Reviewer #3 (Remarks to the Author):

Thank you for the opportunity to review Wu et al.'s manuscript.

The authors perform a metagenomics analysis of a large cohort accompanied by deep phenotyping information. As a starting point, they perform an important replication of previous work finding that poor dietary quality is associated with degree of autism symptoms and measures of microbiome diversity. In this well-powered dataset, they also find associations between poor dietary quality and gastrointestinal complications. The authors then go further, performing a variety of following analyses to characterise dietary-microbiome relationships.

A major strength of the paper is the depth of phenotyping information collected, particularly with respect to dietary measures, and how the authors have carefully explored potential confounders. I particularly like the data-driven analyses that have been performed in Section 2.2.

My questions largely relate to the logical flow of the paper which I believe could be improved to take the reader along on the journey. I often feel unsure what the “deep-dive” analyses (e.g., Section 2.3, 2.4) are being directed by, which can give the impression of cherry-picking (even if it is not). I also believe that the analyses need more statistical testing (rather than the largely qualitative presentation of results at current), plus technical and specific details to be included in the main text (examples below).

Thanks Yap. We much appreciate your thoughtful feedback and constructive suggestions. We would like to address all the comments and response one by one, particularly to enhance the clarity of our deep-dive sections.

Major points:

- For Table 1, please justify why tertiles are used (and not quartiles, quintiles, etc.)? It would be better still if there were known cutoffs for CCDI (rather than being driven by the current data) that you could use to divide. Were tertiles used throughout the analysis (as is suggested in Methods line 403), or just for Figure 1? Please clarify – I believe that CCDI should be kept as a continuous variable in analyses or information is lost unnecessarily.

Response: Thank you for your comments. The choice to use tertiles in Table 1, rather than quartiles or quintiles, is primarily driven by the study sample size, offering a practical approach to illustrate various characteristics and phenotypic information while maintaining enough participants in each group for meaningful comparisons. Since current literature on CCDI has not proven well-defined cutoffs, we believe tertiles-based clarification is supported by many literature^{1,2}. The most important feature of Table 1 is that it shows a significant association between the CCDI and ASD status itself when comparing with demographic factors. We would like to clarify in the main text that the remainder of our analyses retained CCDI as a continuous variable, which has been stated in the revision to improve clarity (line 98-100, line 368-369).

Line 98: A key finding was that the CCDI, among various continuous dietary indices, had a robust association with Shannon diversity that was specific to the ASD group

Line 368: To comprehensively evaluate dietary associations with the gut microbiome, we treated multiple dietary metrics as continuous measures, including indices (CCDI, AHEI, DII,

sulfur-diet score, HFD index, vitamin insufficiency, and mineral insufficiency), individual nutrient and food additive intakes, and food group consumptions—collectively constituting a holistic dietary profiling.

[1] Duan, R., Qiao, T., Chen, Y. et al. The overall diet quality in childhood is prospectively associated with the timing of puberty. *Eur J Nutr.* 2021. 60, 2423–2434. Doi: 10.1007/s00394-020-02425-8

[2] Meng L, Wang Y, Li T, et al. Dietary diversity and food variety in Chinese children aged 3-17 years: are they negatively associated with dietary micronutrient inadequacy? *Nutrients.* 2018. 10(11):1674. doi: 10.3390/nu10111674.

• Given there are various systematic differences shown in Table 1, please articulate what covariates have been included in each analysis, including in the main text.

Response: We thank the reviewer for this critical comment. We have now explicitly stated the included covariates in the main text and Methods section (line 98-102, 126-127, 457-458). Age, sex, and GI conditions were adjusted in all analyses examining microbial species and dietary metrics. Additional variables—including BMI, Bristol Stool Chart, and autistic behaviors—were further adjusted for in the linear regression models evaluating associations between dietary indices and microbial features.

Line 98: A key finding was that the CCDI, among various continuous dietary indices, had a robust association with Shannon diversity that was specific to the ASD group (p for Kendall correlation = 0.049; p for Pearson correlation = 0.006; p for linear regression model adjusted by age, sex, Bristol Stool Chart, GI conditions, and autistic symptoms = 0.010).

Line 126: These associations were analyzed using MaAsLin2, adjusted for the age, sex, and GI conditions that identified above.

Line 457: and generalized regression model with adjustments of age, sex, body mass index, Bristol Stool Chart, GI conditions, and autistic symptoms.

• It is not immediately clear to me how the specific dietary measures shown in Fig.2A and Fig.2B were chosen. Fig. 2B in particular seems to be a subset of Fig.2A. How were these chosen? Were the dietary variables in Fig.2A also a subset and if so how were they chosen?

Response: We thank the reviewer for raising this important point and apologize for any confusion caused by our initial description. To clarify, all correlation analyses between diet and microbial species were performed using the full available dataset, for which complete dietary information was available for all participants, as described in the Methods. The values $n = 28$, $n = 27$, and $n = 18$ referred to the number of microbial species that were significantly associated with dietary components—not the sample size of the study population. We have clarified this in the results (line 148-156) to prevent misunderstanding.

Line 148: The breadth of diet-microbe associations varied by dietary variable, showing stronger connections in ASD compared to their non-ASD peers, who exhibited negligible linkages. The dietary factors that exerted the most substantial effects on microbial species—across nutritional components, dietary indices, and food groups—were protein intake (28 significant associations in total), CCDI scores (27 in total), and bean consumption (18 in total) (Fig. 2B; Supplementary Fig. 4B). By looking into specific nutrients in relation to

microbial shifts, starch and magnesium emerged as the nutrients with the second-highest number of significant associations after protein intake, exhibiting 27 and 20 significant associations, respectively (Fig. 2B).

• For the Fig.2B analysis, is it possible that the difference in classification performance could be due to there being ~100 more people in the ASD group vs TD?

Response: Thank you for this comment. We fully acknowledge the important point regarding the influence of sample size on statistical power. To further substantiate our findings, we conducted a sensitivity analysis using a 1:1-matched ASD subgroup (n = 356) with an equal number of non-ASD participants (Supplementary Table S3, line 104-106). Notably, even with reduced sample size, CCDI showed strengthened associations with microbial features in the ASD group, further supporting the robustness of our initial observations. We have also incorporated the descriptions into the main text, accompanied by a discussion of the potential limitations they may introduce (line 303-307).

Line 104: The findings remained consistent in the sensitivity analysis conducted after excluding samples in accordance with the size and features of non-ASD (Supplementary Table S3).

Line 303: (i) The cross-sectional design precludes causal inference, and the discrepant sample sizes between ASD and non-ASD groups may influence statistics. Nevertheless, sensitivity analyses using matched subgroup sizes consistently confirmed the distinct diet-microbiome associations observed in ASD.

• In Section 2.3 Magnesium/protein enrichment and starch depletion in ASD – I am unclear in what way the protein, starch and magnesium associations are “prominent” (at least when going off Figure 2 where these variables do not stand out). Is this shown in a plot or table somewhere? If so, it would be helpful if this is referenced (and also should probably be in a main text figure). Without this clarity, I feel I am missing the relevance of this section.

Response: Thank you for this comment and for highlighting the need for greater clarity regarding the selection of magnesium/protein enrichment and starch depletion for section 2.3. These specific dietary factors were prioritized based on our initial screening of diet-microbiome associations, as they demonstrated the most substantial influence on the gut microbiota (the greatest number of significant associations with microbial species). We have amended Figure 2B to clearly highlight the numbers of microbial species that were significantly associated with each dietary variable. In addition, we have revised the Results section (line 150–156) to more explicitly state that magnesium, protein, and starch were selected based on the strength and number of their associations with microbial features, and to clarify their contextual importance within central energy-metabolism pathways. These modifications ensure that the rationale for focusing on these particular nutrients and visually supported within the manuscript.

Line 150: The dietary factors that exerted the most substantial effects on microbial species—across nutritional components, dietary indices, and food groups—were protein intake (28 significant associations in total), CCDI scores (27 in total), and bean consumption (18 in total) (Fig. 2B; Supplementary Fig. 4B). By looking into specific nutrients in relation to

microbial shifts, starch and magnesium emerged as the nutrients with the second-highest number of significant associations after protein intake, exhibiting 27 and 20 significant associations, respectively (Fig. 2B).

- Similarly in Section 2.4, please point to what analysis led you to choose to focus on emulsifier exposures? Again, I think this should be in a main text figure.

Response: Thank you for this question, which allows us to clarify the rationale behind our focus on dietary emulsifiers. Our decision was motivated by a convergence of prior evidence and our own analytical findings: First, as noted in the introduction (line 67-69), emulsifiers are widely recognized in the literature as having a potent and direct impact on gut, making them a target for investigation in the context of ASD. Second, our own data supported this focus. As shown in Figure 1, emulsifier exposure emerged as a notable contributor to the variance of microbial compositions. More importantly, the nature of its association with microbial features differed markedly between ASD and non-ASD children (Supplementary Fig. 1C). This suggests the presence of ASD may amplify susceptibility to the microbial perturbations linked to emulsifiers. Finally, the predictive modeling results presented in Figure 2B provided additional support. The relatively high accuracy with which gut microbial profiles could predict emulsifier intake implies the existence of a strong and reproducible microbial signature associated with this exposure. We have now clarified the rationale before starting off the Section 2.4 (line 199-201).

Line 199: Of particular interest were dietary emulsifiers, given their established potential to perturb gut microbiota and their notable contribution to the variance explained in microbial composition (Fig. 1D and Fig 2B).

- In Section 2.4, it would be help logical flow to prime the reader more to the relevance of network analyses and why you have used them.

Response: We appreciate your suggestion. To improve logical flow and prime the reader for the relevance of the subsequent analysis, we have added an introductory sentence at the beginning of Section 2.4 (line 199-201). This addition clarifies that although dietary emulsifiers showed limited broad associations with individual microbial taxa, their significant contribution to the overall variance in microbial composition (Fig. 1D) — combined with prior evidence of their biofilm-disrupting and pro-inflammatory effects — motivated us to examine their influence on microbial co-occurrence network (<https://onlinelibrary.wiley.com/doi/10.1111/all.15825> <https://www.nature.com/articles/s41467-025-62397-3>).

Line 199: Of particular interest were dietary emulsifiers, given their established potential to perturb gut microbiota and their notable contribution to the variance explained in microbial composition (Fig. 1D and Fig 2B).

- In Section 2.4, line 165, “dysbiosis” is mentioned. But what is the definition of dysbiosis here? Probably better to mention as ecological instability if some technical definition of dysbiosis is not met.

Response: Thank you for highlighting this point. In response, we have revised the wording as "ecological instability" (line 208) to better capture the intended meaning without confusion on technical-defined "dysbiosis" that may cause.

- In Section 2.4, please provide more specific metrics of the network differences with statistical testing if possible. At currently, the main text is entirely qualitative which I do not think is sufficient, and I believe Table 2 is unreferenced. Eg. I am not very familiar with network analyses (but the audience may also not be) but I am not immediately convinced that there is a difference between the ASD vs nonASD networks. Eg. "Neurotypical children, in contrast, maintained stable network architecture and functional resilience across emulsifier exposure levels". Please provide statistical tests to support such claims.

Response: Thank you very much for your valuable suggestions. While the topological properties we analyzed (e.g., connectivity, modularity) are well-established metrics for assessing ecological network stability¹, we agree that direct statistical comparisons between groups are essential to robustly support our conclusions. To address this point, we have now performed a Bootstrap Test to quantitatively compare network metrics between the ASD and non-ASD groups, thereby providing statistical evidence for the observed structural differences (described in Methods, line 484–489). Additionally, our original data on simulation attacks plots (figure 4D-E) also supported the stability of network architecture since random or targeted node removals could significantly disrupted network integrity in ASD compared to non-ASD. We have revised the wordings and made the conclusions more readable in the main text (line 207-212).

Line 207: Stratifying children by emulsifier exposure levels (categorized into tertiles) revealed stark network ecological instability exclusively in ASD. To ensure statistical robustness, we performed bootstrap analyses (n = 1000 iterations) for each network metric. In the highest exposure tertile, we observed fragmented microbial connectivity, marked by significant reductions in network degree, number of edges, and average degree (bias-corrected p-values < 0.01)

Line 484: The topological features of the co-occurrence networks were generated by the iGraph package (v4.4.1) and visualized with node sizes proportional to relative abundances. We implemented a nonparametric bootstrap procedure and computed bias-corrected p-values for the statistical differences of each topological feature, with letters indicating significance after multiple comparisons (p < 0.05).

[1] Kajihara, K.T., Hynson, N.A. Networks as tools for defining emergent properties of microbiomes and their stability. *Microbiome*. 2024. 12, 184. doi: 10.1186/s40168-024-01868-z

- In Section 2.4, how is a keystone species defined?

Response: We thank the reviewer for their insightful feedback regarding the use of the term "keystone species." We acknowledge that our initial usage was methodologically imprecise¹. In our study, we intended to highlight taxa that act as highly connected hubs—demonstrating high degree centrality within the co-occurrence network—rather than to invoke the stricter ecological definition often associated with the term. To prevent any potential confusion, we have revised the wording in the manuscript (line 216–219) to describe these taxa.

Line 216: We identified several taxa (e.g., *Enterocloster bolteae*, *Hungatella hathewayi*, and *Clostridium symbiosum*) as potential keystone species due to their high connecting degrees in the interaction network,

[1] Boyse, E., Robinson, K.P., Carr, I.M., et al. Inferring species interactions from co-occurrence networks with environmental DNA metabarcoding data in a coastal marine food web. *Mol Ecol.* 2025. 34: e17701. doi: 10.1111/mec.17701

• For Figure 4 A, what is the x-axis label? For B/C, I'd suggest putting ASD and nonASD labels on the plots to make it easier to understand. What does "high exposures" mean here? For D/E, I would suggest including the nonASD results from SF6, if a key claim the authors are making is that ASD and nonASD networks are fundamentally different.

Response: We thank the reviewer for their suggestions. Regarding Figure 4A, we have updated the figure caption to explicitly indicate that the x-axis represents correlation coefficients derived from MaAsLin2 analyses.

In response to the query concerning the term "high exposure," we have added clarification in the Results section (line 207–210). As standardized limits for emulsifier intake are lacking, we categorized the overall study population into tertiles based on estimated dietary emulsifier consumption, with the highest tertile designated as the "high exposure" group. This methodology was previously detailed in the Table 2 legend. Furthermore, we appreciate the suggestion to incorporate comparative findings from the non-ASD group. We have now expanded the Results to include relevant comparisons (Fig. 4).

Line 207: Stratifying children by emulsifier exposure levels (categorized into tertiles) revealed stark network ecological instability exclusively in ASD. To ensure statistical robustness, we performed bootstrap analyses (n = 1000 iterations) for each network metric. In the highest exposure tertile, we observed fragmented microbial connectivity, Fig. 4

- For Section 2.5, I do not know what specific analyses have actually been performed. Could this please be clarified?

Response: We appreciate your comment and request for clarification on the specific analyses previously performed. The primary goal of this section is to align our findings with the previously published study (<https://www.nature.com/articles/s41564-024-01739-1>), emphasizing that different medications and nutritional supplements did not substantially influence on our previously identified biomarkers (species, genes and pathways) that demonstrated stability for ASD diagnosis. This supports the notion that the interactions between food and microbial communities are independent of these identified biomarkers. We also propose that future research should explore whether these biomarkers impact the overall microbiome community niche, which may, in turn, influence the microbiome-diet interactions. We believe this perspective opens up exciting avenues for further microbiome-targeted strategies. Thank you for the opportunity to elaborate on this important aspect of our study (line 327-329).

Line 327: Whether those ASD-specific biomarkers influence the overall microbiome community niche, which may in turn affect microbiome-diet interactions, opens up exciting avenues for exploration.

- In general, it would be helpful for the reader if the main text included some more details of methods than how the manuscript is currently written. Otherwise, the reader is required to flip back and forth between Methods and Results to make a critical assessment. Eg. For the analysis resulting in Fig.1D, could the authors please specify the model/analysis used in the main text rather than just the Methods (I believe this is PERMANOVA). For the Fig.2B analysis, could the authors please elaborate beyond “machine learning”. Etc.

Response: Thank you for your constructive suggestions on including more methodological detail in the main text. We have revised the manuscript accordingly. We have now explicitly stated in the main text that the statistical analysis performed was a PERMANOVA (Adonis test) based on Bray-Curtis dissimilarities (line 110-112). Regarding the machine learning analysis in Fig. 2B: We have elaborated in the Results section to specify that we used a Random Forest classifier (line 158-162). Other findings with necessary methods explanations needed have also been added in the main text to improve the reader's experience and the clarity of our findings (line 95-98, 126, 208-210).

Line 95: Consistent with prior studies, gut microbiome alpha diversity (Shannon index, $p = 0.024$; observed richness, $p = 0.003$) and dysbiosis score (Dissimilarities-based evaluation as detailed in the Methods, $p = 0.001$) differed significantly between children with ASD and those without

Line 110: To quantify the influence of host factors on gut microbiota composition, we performed a multivariate PERMANOVA analysis through Bray-Curtis distances.

Line 158: using two complementary machine-learning approaches, each trained on relative microbial abundances to predict each dietary variable quantified by correlation coefficients (ρ) for regression and area-under-the-curves (AUCs) for classification (Methods). The random forest regression models demonstrated diet-associated species outperformed functional pathway-based predictions across 100 training/testing folds,

Line 208: To ensure statistical robustness, we performed bootstrap analyses (n = 1000 iterations) for each network metric.

Minor points:

- Figure 2B could be improved by having the rectangle/circle described within the figure key (rather than just the legend).
- Is there a reason why carrageenan is abbreviated to CRN? It would make it easier to read if spelt out in full and wouldn't increase word count.
- There seems to be mention of both non-ASD and TD. Would suggest consistency unless this means something distinct.
- Reference 7 is cited as Johnson and Howell 2021 Cell Metabolism. However, this is commentary rather than the original primary analysis article which is Yap et al. 2021 Cell – I would suggest referencing the latter article (which to be transparent, I acknowledge is mine) as it was the first to empirically support dietary hypotheses of microbiome differences in autism.

Response: Thank you for your thoughtful comments and suggestions. Below are our responses to your specific points: To enhance clarity, we have polished Figure 2B to incorporate the change you suggested.

We appreciate your feedback regarding the abbreviation "CRN" for carrageenan and have spelled it out in full throughout the manuscript.

Regarding consistency in terminology, thank you for pointing out the mention of both non-ASD and TD (typically developing). We have replaced TD with non-ASD and ensured consistency in our terminology.

We appreciate your point regarding the Reference citation and have replaced the citation. Thank you very much for your significant contributions to this research field.

Fig. 2

POINT-BY-POINT REPLY TO EDITORS AND REVIEWERS

Dear reviewers,

We are deeply grateful to the reviewers for their continued engagement and for providing additional insightful feedback, which has further helped us strengthen our work. We have provided a point-by-point response to the reviewers' comments, together with a tracked and clean version of the revised manuscript. All changes are highlighted in yellow.

We appreciate the opportunity to resubmit and hope you find this version acceptable for publication.

Yours sincerely
Siew Ng on behalf of co-authors

Reviewers' Comments:

Reviewer #1:

Remarks to the Author:

The authors have done an excellent job with this revision. The findings and manuscript are much stronger having incorporated responses to all of the critiques. The results from their newly incorporated mediation analysis are particularly interesting.

Response: We are grateful for the reviewer's feedback and are delighted that they find the new mediation analysis interesting; the previous suggestions were instrumental in this improvement.

Reviewer #3:

Remarks to the Author:

1. Thank you very much for the comprehensive replies and additional analyses completed by Wu et al. My major remaining concern (apologies for only identifying one instance of this in my last review) relate to the validity of running separate analyses for ASD and nonASD groups, and then inferring differences based on a direct comparison of p-values between the groups. In essence this is a subgroup analysis of which an overview of pitfalls is provided here: <https://www.nejm.org/doi/10.1056/NEJMSr077003>. The p-value differences identified in these analyses may simply relate to differences in power between the groups, and the coefficients could just relate to different spread of data between groups. The correct analysis needed to identify whether there truly is a significantly different diet/microbiome relationship between the groups is to run a regression with an interaction term. Eg. $\text{microbe} \sim \text{diet} + \text{group} + \text{diet} * \text{group} + \text{covariates}$. If the diet*group term is significant, then you can conclude that the groups have innately differently diet-microbiome associations. These analyses apply for all cases in which ASD and nonASD groups have been compared as subgroups.

Response: We appreciate the reviewer for highlighting this limitation in our initial approach and for proposing the explicit solution. We acknowledge the concern on limited sample size in the between-group analysis and have now performed a comprehensive re-analysis of all diet-microbiome associations using the interaction model (microbe ~ diet + group + diet*group + covariates) to examine the significance of interaction effect (line 142-147, line 186-191). We have now added a new supplementary table S6 and Figure S4 detailing the results (coefficient, adjusted q-value) for all significant interaction terms. Supplementary Table S9 also demonstrated significant diet-by-ASD interaction for certain microbial species, especially *Megamonas funiformis*, which was consistently increased in the ASD group with higher CCDI score and high magnesium intake. These results were updated in the Section 2.3. We hope this new analysis has substantially strengthened our conclusions.

Given that our primary objective was to identify ASD-specific associations that are clinically relevant for this group, we did not prioritize examination of general diet-microbiome associations. Considering the distinct dietary patterns between ASD and non-ASD, pooled data with different spreads may distort associations due to between-group clustering, potentially obscuring associations that are unique to the ASD. To address the concern about sample size disparity, we conducted a 1:1 matched sensitivity analysis (n=356 per group, Supplementary Table S2-3, Table S9) to confirm that diet-microbiome associations remain significant in the ASD group.

Below are details for revision throughout the manuscript:

Table S5: Sensitivity analysis for the diet-microbiome associations for autism spectrum disorder (ASD)-specific species in the 1:1 matched-ASD group (n=356)

Table S6: Effect sizes and adjusted q-values for the diet-microbiome associations incorporated with the interaction term (diet*ASD) across the study population

Table S9: Sensitivity analysis for the alterations of microbial species associated with diet, and diet-by-autism spectrum disorder (ASD) interaction after 1:1-sample size matching. The low intake group is used as the reference compared to the high intake group.

Line 142: These associations were replicated in the ASD group using a sample-size matched sensitivity analysis (Supplementary Table S5). Consistently, the population-wide regression model with an interaction term (ASD*diet) indicated that dietary indices showed substantial interactions with ASD status in relation to the abundance of *L. amygdalina* and *Anaerostipes hadrus* (qval < 0.2, Supplementary Fig. 4; Supplementary Table S6). Notably, decreased abundance of *L. amygdalina* was specifically correlated with insufficient vitamin and mineral intake in the ASD group (|coef| > 0.5 and qval < 0.1), while *Massilioclostridium coli* and *Escherichia coli* exhibited inverse correlations with these dietary metrics (Supplementary Table S4 and Table S5).

Line 186: These trends attenuated in neurotypical peers (Supplementary Fig. 7A-D; Table S8), with minimal overlap in microbial responses—a contrast that persisted against a sample-size-matched ASD cohort (Supplementary Table S9). The ASD-specific increase of *M. funiformis* in response to healthy dietary patterns collectively demonstrated the diet-by-ASD interaction (p for interaction < 0.05, Supplementary Table S9).

Line 320: (i) The cross-sectional design precludes causal inference, and the discrepant sample sizes between ASD and non-ASD groups may influence statistics. Nevertheless, sensitivity analyses using matched subgroup sizes consistently confirmed the distinct diet-microbiome

associations observed in ASD and, meanwhile, cemented an interaction model to support an ASD-specific microbial response.

Line 490: To evaluate whether diet-microbiome associations differed between groups, we fitted a regression model with an interaction term (microbe ~ diet + ASD + ASD*diet + age + sex + GI conditions).

Supplementary Fig. 4: Interaction effects of autism spectrum disorder (ASD) status on diet-microbiome associations. Heatmap displays the identified diet-associated microbial species and their association coefficients with dietary metrics. Coefficients derived from the pooled sample (children with and without ASD) are indicated by circles, and significant coefficients for the ASD-by-diet interaction term are indicated by rectangular. Significant interactions (qval < 0.2) are marked with “+” or “-” corresponding to the effect direction (see Supplementary Table S5 for details). Analyses were adjusted for age, sex, and gastrointestinal conditions using MaAsLin2.

2. Relatedly, I appreciate the inclusion of Supplementary Table 3, but for it to be interpretable, there needs to be a side-by-side comparison of 1) full-ASD group coefficients, 2) subset n=356 ASD group coefficients, 3) non-ASD group coefficients. It also needs to be expanded (probably as separate Supplementary Tables) as sensitivity analyses to stress-test other key results (eg. Firmicutes SGB4348, M funiformis, magnesium/protein-microbiome results, ideally the network results as well).

Response: We sincerely thank the reviewer for this additional suggestion. To address it we have expanded supplementary Table S2 and S3 to include results from both the non-ASD group and the general population, thereby improving the interpretability and comparability of the data (line 102-106). This revision allowed readers to clearly observe that the correlation between

healthy dietary index (CCDI) and gut dysbiosis remained more pronounced in the sample size-matched ASD group compared to the non-ASD. We have also extended sensitivity analyses to key findings (Supplementary Tables S6 and S9). *Firmicutes SGB4348* and *M. funiformis* were consistently associated with healthy dietary indices in the sample size-matched ASD cohort ($n = 356$, line 189-191). Furthermore, Supplementary Table S9 reinforced that *M. funiformis* was substantially influenced by a diet-by-ASD interaction. These updates have been summarized in the revised Results section. Regarding the network results, since all children were stratified into tertiles without separating ASD and non-ASD, we believe this could ensure a fair comparison. Below are details for revision throughout the manuscript:

Table S2: Correlations between various dietary indices and gut microbial features in the general population and those with or without autism spectrum disorder (ASD)

Table S3: Sensitivity analysis of diet–microbiome correlations in a 1:1 matched cohort of children with and without autism spectrum disorder (ASD)

Table S6: Effect sizes and adjusted q-values for the diet-microbiome associations incorporated with the interaction term (diet*ASD) across the study population

Table S9: Sensitivity analysis for the alterations of microbial species associated with diet, and diet-by-autism spectrum disorder (ASD) interaction after 1:1-sample size matching. The low intake group is used as the reference compared to the high intake group

Line 102: In contrast, this association was markedly attenuated in non-autistic peers (Fig. 1C; Supplementary Table S2-3; Supplementary Fig. 1B-C). The findings remained consistent in the sensitivity analysis conducted after excluding samples in accordance with the size and features of non-ASD (Supplementary Table S3).

Line 142: These associations were replicated in the ASD group using a sample-size matched sensitivity analysis (Supplementary Table S5). Consistently, the population-wide regression model with an interaction term (ASD*diet) indicated that dietary indices showed substantial interactions with ASD status in relation to the abundance of *L. amygdalina* and *Anaerostipes hadrus* ($q_{val} < 0.2$, Supplementary Fig. 4; Supplementary Table S6). Notably, decreased abundance of *L. amygdalina* was specifically correlated with insufficient vitamin and mineral intake in the ASD group ($|coef| > 0.5$ and $q_{val} < 0.1$), while *Massilioclostridium coli* and *Escherichia coli* exhibited inverse correlations with these dietary metrics (Supplementary Table S4 and Table S5).

Line 186: These trends attenuated in neurotypical peers (Supplementary Fig. 7A-D; Table S8), with minimal overlap in microbial responses—a contrast that persisted against a sample-size-matched ASD cohort (Supplementary Table S9). The ASD-specific increase of *M. funiformis* in response to healthy dietary patterns collectively demonstrated the diet-by-ASD interaction (p for interaction < 0.05 , Supplementary Table S9).

Specific comments

3. Paragraph starting line 94: could you please include coefficients and standard errors (currently just p-values)

Response: We thank the reviewer for pointing this out. We have revised the paragraph to include the requested coefficients and standard errors for all reported associations (line 94). This addition provides greater transparency regarding the effect sizes and the precision of our estimates.

Below are details for revision throughout the manuscript:

Line 94: Consistent with prior studies, gut microbiome alpha diversity (Shannon index: coef = 0.07, standard error [SE] = 0.03, p = 0.024; observed richness: coef = 9.05, SE = 3.49, p = 0.003) and dysbiosis score (Dissimilarities-based evaluation as detailed in the Methods, coef = -0.02, SE = 0.004, p < 0.001) differed significantly between children with ASD and those without.

4. Section 2.2: Apologies I missed this in the initial review – my (now) understanding of the analysis in this section is that the authors performed analyses within the ASD and non-ASD groups separately and then compare the q-value between the groups to identify “ASD-specific diet-microbiome interactions”. However, to me, a disparity in q-value seems more likely to reflect a difference in power between the groups: the ASD group is larger and also has a greater spread of results (which would make the |coefficient| larger). Thus, I believe the correct analysis is to run the analysis with ASD+nonASD together and including ASD/non-ASD status as an explanatory variable. Then I suggest reporting the coefficient for ASD/nonASD status and identifying ASD-specific diet-microbiome interactions where the q-value for the ASD/non-ASD term meets threshold (eg. line 134-146). I would expect these results to add more robustness to the idea that ASD and non-ASD diet-microbiome associations are in some way distinct.

Response: We thank the reviewer for this critical insight and for proposing a more statistically robust analytical approach. As suggested in response to the aforementioned subgroup pitfalls, we have added a comprehensive re-analysis by including the diet * ASD interaction term across the board (line 142-147, line 186-191, line 324-325). The results, which confirm several significant ASD-specific diet-microbiome interactions, are now reported in the revised manuscript (Supplementary Table S6). Considering sample-size confounding the results, to further support microbial changes in response to diet were ASD-specific, we conducted sample size-matched sensitivity analyses (Supplementary table S2-3, Table S5, Table S9). The key finding on *M. funiformis* was reinforced by a significant diet-by-ASD interaction (p for interaction <0.05, Supplementary Table S9). We believe this new analysis has substantially strengthened our conclusions regarding the distinct nature of these associations in ASD.

Below are details for revision throughout the manuscript:

Table S2: Correlations between various dietary indices and gut microbial features in the general population and those with or without autism spectrum disorder (ASD)

Table S3: Sensitivity analysis of diet–microbiome correlations in a 1:1 matched cohort of children with and without autism spectrum disorder (ASD)

Table S6: Effect sizes and adjusted q-values for the diet-microbiome associations incorporated with the interaction term (diet*ASD) across the study population

Table S9: Sensitivity analysis for the alterations of microbial species associated with diet, and diet-by-autism spectrum disorder (ASD) interaction after 1:1-sample size matching. The low intake group is used as the reference compared to the high intake group.

Line 142: These associations were replicated in the ASD group using a sample-size matched sensitivity analysis (Supplementary Table S5). Consistently, the population-wide regression model with an interaction term (ASD*diet) indicated that dietary indices showed substantial interactions with ASD status in relation to the abundance of *L. amygdalina* and *Anaerostipes hadrus* (qval < 0.2, Supplementary Fig. 4; Supplementary Table S6). Notably, decreased

abundance of *L. amygdalina* was specifically correlated with insufficient vitamin and mineral intake in the ASD group ($|\text{coef}| > 0.5$ and $\text{qval} < 0.1$), while *Massilioclostridium coli* and *Escherichia coli* exhibited inverse correlations with these dietary metrics (Supplementary Table S4 and Table S5).

Line 186: These trends attenuated in neurotypical peers (Supplementary Fig. 7A-D; Table S8), with minimal overlap in microbial responses—a contrast that persisted against a sample-size-matched ASD cohort (Supplementary Table S9). The ASD-specific increase of *M. funiformis* in response to healthy dietary patterns collectively demonstrated the diet-by-ASD interaction (p for interaction < 0.05 , Supplementary Table S9).

Line 320: (i) The cross-sectional design precludes causal inference, and the discrepant sample sizes between ASD and non-ASD groups may influence statistics. Nevertheless, sensitivity analyses using matched subgroup sizes consistently confirmed the distinct diet-microbiome associations observed in ASD and, meanwhile, cemented an interaction model to support an ASD-specific microbial response.

5. Line 124: It would be helpful to briefly justify why Bristol Stool Chart was not also included as a covariate

Response: We appreciate the reviewer's suggestion. The Bristol Stool Chart was not included as a separate covariate because it captures information that is largely encompassed by the broader GI symptom score already in our model (Figure 1D, line 114-115). PERMANOVA analysis indicated that the contribution of GI symptoms was more important than the Bristol Stool Chart. Including both would represent a conceptual and statistical overlap. We have incorporated a brief explanation regarding the covariate selection in the Methods section.

Below are details for revision throughout the manuscript:

Line 488: The latter was selected over the Bristol Stool Chart due to its stronger association with overall microbial community variation.

6. Line 139: “The decreased abundance of *F.SGB4348* also indicated stronger correlations influenced by insufficient vitamin and mineral intake remarkably observed in ASD” – I’m not sure what “remarkably observed” means here

Response: We thank the reviewer for prompting us to clarify this point. Our original phrasing intended to highlight that the correlation between reduced *F.SGB4348* abundance and insufficient vitamin/mineral intake was a distinctive feature specific to ASD. However, following the reviewer's suggestion to implement an interaction model and sensitivity analyses, we have re-evaluated our findings. This led us to amend the manuscript to feature the strong and consistent result for *L. amygdalina*, although the *F.SGB4348* result remained significant.

Below are details for revision throughout the manuscript:

Line 147: Notably, decreased abundance of *L. amygdalina* was specifically correlated with insufficient vitamin and mineral intake in the ASD group ($|\text{coef}| > 0.5$ and $\text{qval} < 0.1$), while *Massilioclostridium coli* and *Escherichia coli* exhibited inverse correlations with these dietary metrics (Supplementary Table S4 and Table S5).

7. Reply to Figure 2B analysis classification performance: Thank you for this information. It would also help in the manuscript to state how you chose the matched subgroup.

Response: We appreciate the reviewer's valuable feedback on this point. The matched subgroup was selected using the MatchIt package in R. A 1:1 nearest-neighbor matching algorithm was applied based on the covariates of age, sex, and GI condition status. This procedure ensured that the ASD and non-ASD groups were balanced on these key demographic factors, thereby strengthening the robustness of our comparative analyses. We have now clarified in the manuscript:

Line 495: To ensure sufficient statistical power for within-group comparisons of ASD and non-ASD participants, we applied a 1:1 nearest-neighbor matching algorithm using the MatchIt package in R, matching on age, sex, and GI conditions.

8. Line 149, 167 – “exerted the most substantial effects”, “outsized role in modulating ...” implies causality which I don’t think can be determined from this analysis. Suggest rewording.

Response: We thank the reviewer for this valuable comment and for catching this overstatement. We have carefully revised the text to correct this and ensure our interpretations are precise. The changes have been made as follows:

Line 157: The dietary factors exhibiting the greatest magnitude of associations with microbial species—

Line 174: Notably, emulsifiers—particularly polysorbate-80 and carrageenan—demonstrated stronger associations with microbial features in the ASD group (Fig. 2B; Supplementary Fig. 6).

It revealed that dietary indices exerted a more pronounced association with microbial features in ASD children without GI complications (Supplementary Fig. 11),

Line 342: Overall, our findings position the ASD gut microbiome as a factor both engaged in an interplay with diet and associated with dietary responses,

9. Line 179: Again, unclear whether the attenuated effect size could be a power issue / from there being less spread in the data. Again, suggest analysis with all individuals and extract coefficient for ASD/non-ASD status as an explanatory variable. Then for species associated with both the relevant dietary measures and ASD group status, can do GSEA analysis.

Response: We appreciate the reviewer's emphasis on analytical rigor. After careful consideration, we have prioritized analytical approaches to focus on the core aim: uncovering ASD-specific microbial responses to diet-- a question raising on diet-by-ASD interaction.

As suggested, we have implemented the following in our revised manuscript: We supplemented a unified model including a diet*ASD_status interaction term. We conducted a sensitivity analysis using a 1:1-matched ASD subgroup. This analysis robustly demonstrates that the stronger diet-microbe associations in the ASD group are not merely an artifact of sample size or data spread, but a replicable and clinically relevant finding. While GSEA is a powerful tool for identifying pre-defined gene sets associated with a single phenotype (such as ASD status), its falls outside the scope of our present study. We appreciate this insightful perspective and have incorporated a discussion of it as a valuable direction for future research.

Below are details for revision throughout the manuscript:

Line 104: The findings remained consistent in the sensitivity analysis conducted after excluding

samples in accordance with the size and features of non-ASD (Supplementary Table S3).

Line 142: These associations were replicated in the ASD group using a sample-size matched sensitivity analysis (Supplementary Table S5). Consistently, the population-wide regression model with an interaction term (ASD*diet) indicated that dietary indices showed substantial interactions with ASD status in relation to the abundance of *L. amygdalina* and *Anaerostipes hadrus* ($q_{val} < 0.2$, Supplementary Fig. 4; Supplementary Table S6).

Line 322: Nevertheless, sensitivity analyses using matched subgroup sizes consistently confirmed the distinct diet-microbiome associations observed in ASD and, meanwhile, cemented an interaction model to support an ASD-specific microbial response.

Line 339: (iv) Finally, microbial network analyses warrant mechanistic explorations into microbiome-directed signals and gene-enrichment functions, ultimately requiring validation in gnotobiotic models.

10. Paragraph starting line 190: please add statistics from mediation analysis.

Response: We thank the reviewer for this suggestion. We have now added the detailed statistics from the mediation analysis (including the average causal mediation effect with proportion) to the paragraph (line 202-211).

Line 202: Among the specific nutritional components examined, *Faecalibacterium prausnitzii* exerted a beneficial mediating influence on the associations between protein/zinc intake and social communication (average causal mediation effect [ACME]= -0.125, p for mediation = 0.006, proportion = 24.6%) and hyposensitivity (ACME = -0.004, p = 0.02, 15.3%) (Supplementary Fig. 7; Supplementary Table S8). Notably, *Coprococcus eutactus* emerged as the most prominent mediator, exhibiting the highest proportion of effect for several diet-restricted and repetitive behaviors (RRB) associations, including CCDI (ACME = 0.013, p = 0.02, 15.8%), DII (ACME = -0.098, p = 0.02, 19.9%), and bean consumption (ACME = 0.009, p = 0.02, 35.4%).

11. Line 206: Please clarify whether children were split into tertiles when aggregated together or within the ASD/non-ASD groups. I believe that the former is required to make a fair comparison.

Response: We are grateful to the reviewer for prompting us to clarify this methodological point. As suggested, we have revised the Methods (line 514-516) to explicitly state that children were split into tertiles based on pooled data from the entire cohort. This methodological choice was also stated in the Results section (line 221).

Below are details for revision throughout the manuscript:

Line 221: Characterization of population-wide emulsifier exposure tertiles of the cohort revealed demonstrated ASD-specific microbial network instability,

Line 514: To assess differences in microbial connection network, we compared ASD and non-ASD groups within tertiles of food emulsifier exposure, which were calculated across all participants to establish consistent risk strata.

12. Reply to Section 2.4 “dysbiosis” vs “ecological instability”: please also define exactly what is meant by ecological instability or whatever metric it is you use (apologies my comment before was unclear)

Response: We sincerely thank the reviewer for this clarifying comment and apologize for the lack of precision in our initial description. We have revised the manuscript to clarify that "ecological instability" is operationalized through the comparative analysis of network topology between groups, rather than by a defined cutoff (line 516-519). Specifically, in the revised Section 2.4, we now define it as a state characterized by a breakdown of topological structure, manifesting as a significant reduction in both the average node degree (indicating fewer microbial interactions) and the clustering coefficient (reflecting a loss of modular organization). The Methods section has been updated to cite the basis for these metrics and to describe the bootstrapping approach used for statistical comparison (line 516-519). Additionally, we have recalled the previous Supplementary Fig. 1C, reflecting high exposure to polysorbate-80 was associated with increased dysbiosis in ASD (line 223-225).

Below are details for revision throughout the manuscript:

Line 221: Characterization of population-wide emulsifier exposure tertiles of the cohort revealed demonstrated ASD-specific microbial network instability, evident from significant topological alterations (Table 2) and a positive association between emulsifier levels and dysbiosis score (Supplementary Fig. 1C). To ensure statistical robustness, we performed bootstrap analyses ($n = 1000$ iterations) for each network metric. In the highest exposure tertile, we observed fragmented microbial connectivity, marked by significant reductions in network degree, number of edges, and average degree (bias-corrected p -values < 0.01) (Fig. 4B-C; Table 2; Supplementary Fig. 9A-B).

Line 516: We assessed the significance of differences in topological features—such as reduced average node degree and clustering coefficient, which reflect ecological instability—using a non-parametric bootstrap procedure to compute bias-corrected p -values, with significance letters indicating $p < 0.05$ after multiple comparison adjustment.

Reviewer #4:

Remarks to the Author:

Response: We truly appreciate you sharing your expertise and insightful comments by co-reviewing with other reviewers. We have rigorously addressed all your suggestions. Most notably, we have bolstered the statistical rigor by implementing unified models with interaction terms to formally test our core hypothesis, and we have provided comprehensive sensitivity analyses. Thank you again for your invaluable contribution.

POINT-BY-POINT REPLY TO EDITORS AND REVIEWERS

Dear reviewers,

We wish to express our gratitude to the reviewers for their ongoing guidance and insightful comments, which have been instrumental in enhancing this work. We have provided a point-by-point response to the reviewers' comments, together with a tracked and clean version of the revised manuscript. All changes are highlighted in yellow.

We appreciate the opportunity to resubmit and hope you find this version acceptable for publication.

Yours sincerely

Siew Ng on behalf of co-authors

Reviewers' Comments:

Reviewer #3:

Remarks to the Author:

1. Thank you very much to the authors for the additional analyses that have been performed. It is much appreciated.

Line 102-104: This analysis comparing effects between groups should include both ASD and non-autistic peers in a regression of: $CCDI \sim \text{Shannon diversity} + \text{ASDgroup} + \text{Shannon diversity} * \text{ASDgroup} + \text{covariates}$. You can then claim that the ASDgroup has a significantly stronger association if the interaction term is significant.

Response: Thank you for this insightful comment regarding the use of an interaction model. We have now added this as a supplementary analysis (Table S2), while prioritizing the ASD-specific analysis that guided by the distinct focus of our research question. Our aim was to delineate the structure of diet-microbiome-phenotype relationships specifically within the ASD population, which is more straightforward by describing the relationship network within the ASD group. Finding no statistical significance of interaction may cause by limited sample size, insufficient variation in one or both of the predictor variables to show a clear interaction, or unmodeled nonlinear relationships (<https://besjournals.onlinelibrary.wiley.com/doi/10.1111/2041-210X.13714>). For example, potential correlation between GI conditions and ASD status could inflate variance, making it difficult to disentangle their unique effects and potentially obscuring the significance of their interaction. Therefore, we employed a 1:1 matched cohort and sensitivity analyses to facilitate a more clinically interpretable assessment of findings specific to ASD.

Table S2: Relationships between various dietary indices and gut microbial features specific to autism spectrum disorder (ASD) and the effect modification of ASD across the study population

2. Sections 2.2 and 2.3 (relevant sections: lines 136-152; lines 184-193) Thank you for performing these analyses with the population-wide regression model + interaction term. The Supplementary Tables are very helpful and useful. However, I believe that the order

of analyses should be the other way around. They are currently presented as 1) subgroup analysis □ 2) population analysis. However, I believe it should be 1) population analysis (ie. to test the question “Is there a significant ASDgroup difference in dietary feature X?”) □ 2) subgroup analysis to test the magnitude of the difference.

Response: We appreciate the reviewer's suggestion regarding the order of results. We believe this may help to clarify the primary narrative of our study, while the manuscript was designed with a specific, clinically driven focus: to directly identify and characterize associations that are most salient and relevant within the ASD population. This priority for discovery within the ASD group guided our analytical sequence. Presenting the ASD group analysis first allows us to immediately introduce the reader to the core clinical landscape of our findings—the specific dietary-microbiome relationships that are most pertinent to ASD children. The subsequent population-wide analysis with interaction terms then serves a crucial, but complementary, role. This sequence—from discovery within the target population to confirmation of its specificity—creates a narrative flow that best serves our primary objective. As noted, the limited sample size and other complex factors may attenuate the statistical power to detect interactions, which is precisely why we complemented it with additional, robust methods like 1:1 nearest-neighbor matching analysis.

To enhance the logical coherence of our story, we have revised the main text to better articulate the rationale behind the sequence of our results:

Line 103: In contrast, we did not observe this association in non-autistic peers (Fig. 1C; Supplementary Table S2-3; Supplementary Fig. 1B-C). While the effect modification by ASD status did not attain statistical significance, the consistent pattern of association was verified in sensitivity analysis that excluded samples in accordance with the size and features of non-ASD (Supplementary Table S2-3).

Line 123: Building on the observed connections between diet, food additives, and microbial ecology, we systematically evaluated associations between dietary factors (indices, components, and food groups) and gut microbiome profiles within the ASD population.

Line 133: revealed distinct diet-microbe associations comparing the non-ASD peers when conducting the same analysis procedure.

Line 147: Consistently, the population-wide regression model with an interaction term (diet × ASD) indicated the effect modification of ASD status on dietary associations with the abundance of *L. amygdalina* and *Anaerostipes hadr*

3. Paragraph starting line 203: please indicate whether the p-values reported are with multiple testing correction.

Response: We thank the reviewer for raising this issue. We apologize for the unclear presentation of the results and have revised to clarify that significant p-values reported were adjusted by multiple testing correction (updated in Source Data file, line 213-214).